# Evolving SARS-CoV-2 Vaccines: From Current Solutions to Broad-Spectrum Protection

**DOI:** 10.3390/vaccines13060635

**Published:** 2025-06-12

**Authors:** Rui Qiao, Jiayan Li, Jiami Gong, Yuchen Shao, Jizhen Yu, Yumeng Chen, Yinying Lu, Luxuan Yang, Luanfeng Lin, Zixin Hu, Pengfei Wang, Xiaoyu Zhao, Wenhong Zhang

**Affiliations:** 1Shanghai Sci-Tech Inno Center for Infection & Immunity, National Medical Center for Infectious Diseases, Huashan Hospital, Institute of Infection and Health, Fudan University, Shanghai 200438, China; 2Shanghai Pudong Hospital, Fudan University Pudong Medical Center, State Key Laboratory of Genetics and Development of Complex Phenotypes, MOE Engineering Research Center of Gene Technology, School of Life Sciences, Shanghai Institute of Infectious Disease and Biosecurity, Shanghai Key Laboratory of Oncology Target Discovery and Antibody Drug Development, Fudan University, Shanghai 200438, China; 3Department of Infectious Disease, Mengchao Hepatobiliary Hospital of Fujian Medical University, Fuzhou 350122, China; 4Artificial Intelligence Innovation and Incubation Institute, Fudan University, Shanghai 200438, China; 5Shanghai Academy of Artificial Intelligence for Science, Shanghai 200240, China

**Keywords:** coronavirus, sarbecovirus, vaccines, immune evasion, broadly neutralizing antibodies

## Abstract

The continuous evolution of severe acute respiratory syndrome coronavirus 2 (SARS-CoV-2) and the emergence of variants of concern (VOCs) underscore the critical role of vaccination in pandemic control. These mutations not only enhance viral infectivity but also facilitate immune evasion and diminish vaccine efficacy, necessitating ongoing surveillance and vaccine adaptation. Current SARS-CoV-2 vaccines, including inactivated, live-attenuated, viral vector, protein subunit, virus-like particle, and nucleic acid vaccines, face challenges due to the immune evasion strategies of emerging variants. Moreover, other sarbecoviruses, such as SARS-CoV-1 and SARS-related coronaviruses (SARSr-CoVs) pose a potential risk for future outbreaks. Thus, developing vaccines capable of countering emerging SARS-CoV-2 variants and providing broad protection against multiple sarbecoviruses is imperative. Several innovative vaccine platforms are being investigated to elicit broad-spectrum neutralizing antibody responses, offering protection against both current SARS-CoV-2 variants and other sarbecoviruses. This review presents an updated overview of the key target antigens and therapeutic strategies employed in current SARS-CoV-2 vaccines. Additionally, we summarize ongoing approaches for the development of vaccines targeting infectious sarbecoviruses.

## 1. Introduction

Severe acute respiratory syndrome coronavirus 2 (SARS-CoV-2) was first identified in December 2019 and rapidly spread globally, leading to the COVID-19 pandemic [1,2]. This novel coronavirus primarily causes respiratory symptoms such as fever, cough, and chest discomfort. More importantly, SARS-CoV-2 infection can sporadically lead to acute respiratory distress syndrome (ARDS), multi-organ failure, and even death [3]. As of March 2025, over 777 million confirmed cases and more than 7 million deaths have been reported worldwide (https://data.who.int/ (accessed on 8 June 2025)). Extensive global research has focused on elucidating the structure and pathogenesis of SARS-CoV-2 [4,5,6]. The viral genome encodes four major structural proteins: the spike (S) protein, nucleocapsid (N) protein, membrane (M) protein, and envelope (E) protein (Figure 1A). Vaccine development has concentrated on the full-length S protein and its receptor-binding domain (RBD), as both are crucial for eliciting neutralizing antibody responses (Figure 1B). Additionally, other viral proteins, including the main protease (Mpro), papain-like protease (PLpro), and RNA-dependent RNA polymerase (RdRp), have emerged as promising therapeutic targets [7].

Similarly to other viral infections, SARS-CoV-2 infection activates the body’s immune response, engaging both the innate and adaptive immune systems [8]. The innate immune response is initiated within hours of viral entry, preceding the activation of the adaptive immune system [9]. Viral entry is primarily mediated by surface receptors, with Angiotensin-Converting Enzyme 2 (ACE2) being the primary receptor for SARS-CoV-2, as well as other zoonotic coronaviruses such as SARS-CoV-1 and HCoV-NL63 [10,11]. ACE2 is highly expressed in ciliated secretory cells of the nasal cavity and type II alveolar cells in the lungs, explaining the frequent involvement of both the upper and lower respiratory tracts in SARS-CoV-2 infection [12]. Upon infection, SARS-CoV-2 activates pattern recognition receptors (PRRs), triggering the release of pro-inflammatory cytokines, including IL-6, IL-1β, TNF, IL-12, IFN-β, and IFN-γ [13]. While these cytokines contribute to viral clearance, excessive or dysregulated cytokine release can result in a cytokine storm, leading to severe disease [14].

Adaptive immune responses are antigen-specific and essential for controlling and eliminating viral infections. In this context, CD4^+^ T cells serve critical helper and effector functions, while CD8^+^ T cells act as the primary mediators of cell-mediated immunity, responsible for clearing infected cells [15]. B cells, in turn, produce antibodies that are crucial for humoral immunity. Following symptom onset, levels of SARS-CoV-2-specific IgG and IgM gradually increase in the serum, with seroconversion occurring either simultaneously or sequentially [16]. Most SARS-CoV-2 patients develop neutralizing antibodies (Nabs), and the timing of this response closely correlates with seroconversion, with the S protein being the primary target for these antibodies. Moreover, cell-mediated immunity, a critical component of the adaptive immune response, plays a significant role in viral clearance [9]. Studies have shown that the number of SARS-CoV-2-specific T cells in the lungs correlates with clinical protection [17]. Internal viral proteins, such as the highly conserved nucleocapsid (N) protein, represent ideal vaccine targets for cytotoxic CD8^+^ T cell activation due to their sequence stability. CD8^+^ T cells recognize peptide-HLA complexes with varying affinity, with studies demonstrating that HLA-B7^+^ individuals mount robust responses against the SARS-CoV-2 N protein’s immunodominant SPR epitope (>80% recognition in convalescents). Cross-reactive memory CD8^+^ T cells potentially drive polyfunctional, high-avidity responses to SPR, thereby providing T cell immunity [18]. Meanwhile, adaptive immune responses and immune memory are fundamental to vaccine efficacy.

Building on a comprehensive understanding of the immune system, research has led to the development of various strategies for viral infection prevention, with the creation of effective and safe vaccines being crucial for controlling the COVID-19 pandemic. Significant progress was made in the development of first-generation SARS-CoV-2 vaccines within the first two years of the outbreak. Vaccines such as BNT162b2, mRNA-1273, ZF2001, and BBIBP-CorV played a critical role in early pandemic control. However, the continual emergence of SARS-CoV-2 variants, some harboring mutations in key viral entry regions, has challenged their protective efficacy (Figure 1C). Variants of concern (VOCs) such as B.1.1.7 (Alpha), B.1.351 (Beta), P.1 (Gamma), B.1.617.2 (Delta), and B.1.1529 (Omicron) significantly impacted vaccine effectiveness. More recently, Omicron subvariants, including XBB.1.5, XBB.1.16, EG.5.1, HK.3, and descendants of BA.2, such as JN.1 and KP.3.1.1, have rapidly outcompeted other VOCs to become globally dominant [19]. These circulating variants possess additional mutations that confer substantial or complete resistance to neutralization by nearly all authorized vaccines and antibodies [20,21,22]. First-generation vaccines, particularly BBIBP-CorV, show limited efficacy against early Omicron strains, with recipients displaying minimal neutralizing activity [23]. Similarly, individuals vaccinated with mRNA vaccines, such as mRNA-1273, BNT162b2, or AZD1222, exhibited detectable but significantly reduced neutralizing activity—over 20-fold lower than that observed against the original SARS-CoV-2 [24]. This highlights the urgent need for next-generation SARS-CoV-2 vaccines capable of providing broad protection against both current and future variants.

This review begins by examining the immunogenic characteristics of SARS-CoV-2 structural proteins and summarizes global progress in vaccine development, with a focus on inactivated, live-attenuated, viral vector-based, protein subunit, virus-like particle (VLP), and nucleic acid vaccine platforms. Additionally, we explore strategies to enhance vaccine efficacy in next-generation vaccine development. Understanding these diverse vaccine platforms and optimization approaches is crucial for addressing the challenges posed by emerging SARS-CoV-2 variants and ensuring long-term protection against COVID-19. The insights presented in this review will support ongoing efforts to design more effective vaccines, contribute to global vaccination strategies, and aid in controlling the pandemic while mitigating future outbreaks.

## 2. Immunogenic Features of SARS-CoV-2 Structural Proteins

### 2.1. Full-Length S Protein

As mentioned above, the S protein is crucial for receptor recognition, viral binding, and membrane fusion, making it the primary target for most SARS-CoV-2 vaccines [25]. The full-length S protein is particularly significant, as it retains its native conformation, enhancing immunogenicity by exposing additional epitopes [26]. During infection, the S protein is cleaved by the Furin into S1 and S2 subunits. The pre-fusion conformation of the S protein elicits stronger immunogenicity, making it an ideal vaccine target. Most NAbs recognize epitopes in this conformation, whereas the post-fusion state masks these epitopes, reducing neutralizing antibody production [27]. Of note, the 2P or 6P mutation strategy stabilizes the pre-fusion conformation of the S protein [4]. In addition, mutating the RRAR motif to GSAS at the S1/S2 cleavage site prevents cleavage in host cells, preserving structural integrity and enhancing immune responses [28]. As a result, the full-length S protein has been widely incorporated into various vaccine platforms, including protein subunit vaccines like NVX-CoV2373, which uses recombinant nanoparticle forms of the full-length S protein; mRNA vaccines such as BNT162b2 and mRNA-1273, which encode the full-length S protein; and adenoviral vector vaccines like AZD1222 and Ad26.COV2.S, which deliver genetic instructions for in vivo S protein production. As viruses continue to evolve, vaccines based on the wild-type S protein have become increasingly less effective against emerging variants.

### 2.2. RBD

The S protein exists as a homotrimer and is composed of two functional subunits: S1 and S2. The S1 subunit is further divided into the N-terminal domain (NTD) and the RBD, both essential for viral entry by interacting with host cell receptors. The RBD adopts two distinct conformational states: the closed “down” state and the open “up” state [29]. In the “down” state, the RBD is positioned near the central cavity of the trimer, shielding the receptor-binding region. Conversely, in the “up” state, the receptor-binding region is exposed, allowing interaction with the human ACE2 receptor; however, this state is less stable. Based on these conformational states, antibodies targeting the RBD can be classified into four distinct categories [29,30]. For instance, Class 1 antibodies, such as CB6, bind exclusively to the “up” RBD; Class 2 antibodies, like LY-CoV 555, recognize both the “up” and “down” states; Class 3 antibodies, exemplified by S309, bind outside the ACE2-binding region and interact with both conformations; Class 4 antibodies, such as CR3022, bind both states but do not block the ACE2-RBD interaction.

Classifying antibodies based on their epitope recognition provides critical structural insights for designing antibody cocktails and RBD-based vaccines. These insights have guided vaccine development targeting diverse SARS-CoV-2 variants. For example, Liu et al. [31] demonstrated that rhesus macaques immunized with the RBD-Fc/CF501 vaccine produced broad neutralizing sera effective against XBB.1.5, XBB.1.16, CH.1.1, EG.5, BA.2.86, and JN.1. Despite extensive RBD mutations in variants like JN.1, the original RBD-Fc antigen with CF501 adjuvant elicited strong cross-NAbs. Similarly, An et al. [32] identified key immune escape mutations in the RBD and incorporated them into a modified RBD, which conferred broad protective efficacy against multiple SARS-CoV-2 variants.

### 2.3. NTD

Apart from the RBD, other regions of the S protein, such as the NTD, also contribute to immune responses. The NTD is highly glycosylated, which limits antigenic epitope exposure. Early NTD-targeting antibodies primarily recognized a specific “supersite”, with fewer binding outside this region [33]. While these antibodies do not directly block receptor binding, they can interfere with the viral entry or prevent the S protein from transitioning to its post-fusion conformation. Compared to RBD-specific antibodies, NTD-targeting antibodies generally exhibit weaker neutralizing activity. However, combining NTD-targeting antibodies with those recognizing other epitopes can enhance overall protection. Incorporating the NTD into COVID-19 vaccines can expand the range of neutralizing epitopes, thus improving vaccine efficacy against variants and mitigating immune escape.

### 2.4. S2 Subunit

Another promising target for broad protection is the S2 subunit of the S protein, which contains several critical functional domains. These include the fusion peptide (FP), heptad repeat 1 (HR1), central helix (CH), connector domain (CD), stem helix (SH), heptad repeat 2 (HR2), and the transmembrane anchor (TM). During viral entry, the S2 subunit undergoes a conformational change, with HR1 and HR2 interacting to form a six-helix bundle (6-HB) fusion core [34]. In its pre-fusion conformation, the S2 subunit is shielded by the S1 subunit, making it more conserved. Recent studies have shown that NAbs targeting the FP and SH region provide broad-spectrum neutralization against β-coronaviruses [35,36,37,38]. The conserved nature of the S2 subunit makes it an attractive target for vaccines designed to offer broad protection against both SARS-CoV-2 and other human coronaviruses. For instance, Wang et al. developed a recombinant protein vaccine, HR1LS, based on the HR1, CH, and SH regions of the S2 subunit, and the vaccine was shown to neutralize multiple coronaviruses in vitro [39]. Similarly, Lu et al. designed a recombinant subunit vaccine, HR121, targeting the conserved HR1 domain of the S2 subunit, which provided protection against SARS-CoV-2 in animal models [40].

### 2.5. Other Proteins

In addition to the S protein, the N protein is a key structural component of SARS-CoV-2, sharing 90% homology with the N protein of SARS-CoV [41]. The N protein comprises 419 amino acids and includes an NTD, a C-terminal domain (CT), and three intrinsically disordered regions (IDRs). Although the N protein plays a key role in the formation of ribonucleoprotein complexes (RNPs), its immunogenicity has not been extensively documented. The SARS-CoV-2 N protein contains conserved peptides that bind to human leukocyte antigen (HLA) epitopes for both CD4^+^ and CD8^+^ T cells [42]. This protein also mediates innate immunity through Fc receptors (FcγR)-dependent mechanisms, enhancing T cell activation and phagocytosis during infection [43]. For instance, T cell-mediated immunity is essential for viral clearance in upper respiratory tract tissues, particularly in the nasal mucosa in mice [44]. However, the N protein’s intracellular location limits its recognition by immune cells, preventing the production of NAbs. Moreover, in the absence of S protein, N protein production failed to induce potent serum-neutralizing antibody responses, highlighting its limited immunogenic potential [45]. These findings suggest that further research is needed to fully evaluate the N protein’s potential as a vaccine target.

Beyond the N protein, the M protein is another crucial viral component. The M protein is a transmembrane glycoprotein embedded in the viral membrane, consisting of three transmembrane domains and a conserved cytoplasmic domain that helps regulate the host immune response [46]. Across various sarbecovirus strains, the M protein exhibits a high degree of amino acid sequence conservation, suggesting that immune responses targeting this protein may offer cross-protective benefits. Despite its importance in the viral life cycle, the M protein is rarely used as an immunogen. Notably, only its ectodomain appears capable of eliciting a protective antibody response [47]. For instance, a highly immunogenic peptide, S2M2–30, has been identified within the M protein ectodomain. When conjugated to the keyhole limpet hemocyanin (KLH) carrier protein, S2M2–30 induced robust peptide-specific antibody and cellular immune responses [47]. Similarly, the E protein, which forms an ion channel, can trigger severe inflammatory responses and significantly influence viral pathogenicity and immune evasion [48]. Like the M protein, the E protein is also infrequently utilized as a primary immunogen.

## 3. Development of SARS-CoV-2 Vaccines

The functional characteristics of key spike protein domains directly guide vaccine targeting strategies. The RBD, as the primary interface with ACE2, serves as a precise and effective target for mRNA-based vaccines. In contrast, the strong immunogenicity of the NTD supports the inclusion of the full-length spike protein in inactivated vaccine formulations, ensuring broad epitope coverage, including conformational determinants. In response to the urgent global demand for COVID-19 countermeasures, a diverse array of SARS-CoV-2 vaccine platforms has been developed or is currently under development (Figure 2 and Table 1).

### 3.1. Inactivated Vaccines and Live-Attenuated Vaccines

Inactivated vaccines are the most widely used type, developed by inactivating or killing the disease-causing virus or bacterium using chemicals, heat, or radiation [49]. Unlike vaccines based solely on specific structural proteins of the virus, such as the S protein, inactivated vaccines use the entire virus as an immunogen, which can induce a broader range of antibodies against multiple epitopes [50]. Whole-virus vaccines played a critical role in the early pandemic phase, demonstrating rapid deployability in resource-limited settings and foundational efficacy approaching 70% against the WT strain [51].

**Table 1 vaccines-13-00635-t001:** Summary of SARS-CoV-2 vaccines.

No.	Vaccine Platform	Vaccine Name	Detailed Information	Immunization Route	Developers	Article
1	Inactivated vaccines	CoronaVac	Chemically inactivated SARS-CoV-2 and aluminum hydroxide as an adjuvant	IM	Sinovac Biotech	[52]
2		BBIBP-CorV	Inactivated SARS-CoV-2 against pre-Omicron strains	IM	Sinopharm BIBP	[53]
3		Covaxin	Whole-virion inactivated SARS-CoV-2 vaccine with TLR7/8 agonist adsorbed to alum	IM	Bharat Biotech-ICMR-NIV	[54]
4		QazCovid-in	Kazakhstan SARS-CoV-2 isolated, inactivated with formaldehyde, adjuvanted with alum	IM	RIBSP	[55]
5		VLA2001	β-Propiolactone inactivates virus with CpG 1018 and aluminum hydroxide	IM	Valneva SE	[56]
6		WIV04	Chemically inactivated SARS-CoV-2 WIV04 with aluminum hydroxide adjuvant	IM	WIBP	[57]
7		KCONVAC	19nCoV-CDC-Tan-Strain, chemically inactivated, with alum adjuvant	IM	Kangtai & Minhai, China	[58]
8		BIV1-CovIran	Chemically inactivated SARS-CoV-2 and aluminum hydroxide as an adjuvant	IM	Pasteur Institute of Iran	[59]
9	Live-attenuated vaccine	COVI-VAC	Mimic infection, stimulate immunity, codon deoptimization, enhanced safety	IN	Codagenix	[60]
10		∆3678	Deleted ORF 3, 6, 7, 8; ∆3678 replicates 7500-fold lower than wild-type in airway cultures	IN	DBMB	[61]
11		dCoV	WA/1 strain with sub-optimal codons and deleted furin sites	IN/IM	SIIPL	[62]
12		QazCOVID-Live	Attenuated SARS-CoV-2 via Vero cell passages	IN	RIBSP	[63]
13	Protein-based vaccine	SCTV01E	Tetravalent vaccine with Alpha, Beta, Delta, Omicron BA.1 S-ECD, plus SCT-VA02B	IM	Sinocelltech	[64]
14		NVSI-06-07	Trimeric RBDs from SARS-CoV-2; NVSI-06-07 boosts nAb response	IM	National Vaccine &Serum Institute, NVSI	[65]
15		NVSI-06-09	Trimeric RBD integrates Omicron and variant mutations into a mosaic vaccine			
16		Nuvaxovid (NVX-CoV2373)	Recombinant nanoparticle vaccine with S protein and Matrix-M adjuvant	IM	Novavax	[66]
17		EpiVacCorona	SARS-CoV-2 S protein conjugated to carrier protein, adsorbed on aluminum hydroxide	IM	FSRCVB	[67]
18		Zifivax (ZF2001)	Dimeric RBD with alum adjuvant, 3-dose regimen, robust T cell responses	IM	CAS Microbiology	[68]
19		MVC-COV1901	Utilize CHO cells and contain CpG 1018 and aluminum hydroxide as adjuvants	IM	MVC	[69]
20		Corbevax	Pichia pastoris encodes SARS-CoV-2 RBD, adjuvanted with alum and CpG 1018	IM	Baylor Vaccine Center& Biological E. Limited	[70]
21		CIGB-66 (Abdala)	Pichia pastoris yeast platform encodes SARS-CoV-2 RBD, adjuvanted with alum adjuvant	IM	CIGB	[71]
22		VidPrevtyn Beta	Bivalent vaccine (D614, Beta B.1.351) with GSK AS03 adjuvant	IM	Sanofi &GSK	[72]
23		R-CNP	Nanoparticles with cholera toxin B subunit displaying SARS-CoV-2 RBD, alveoli delivery	IN	Clover Biopharmaceuticals	[73]
24		SCTV01E-2	Recombinant S-ECD protein from Beta, Omicron BA.1, BQ.1.1, XBB.1	IM	Sinocelltech	[74]
25		HR1LS	Target HR1, CH, SH regions, neutralize multiple coronaviruses in vitro	IM	Wang X, et al.	[39]
26	Virus-like particle vaccine	Covifenz	Used plant-derived VLPs, demonstrated efficacy against the Delta and Gamma	IM	Medicago	[75]
27		DVLP	DC-targeting VLP with engineered Sindbis glycoprotein, packaging SARS-CoV-2 Spike mRNA	IM	SJTU	[76]
28	DNA-based vaccine	INO-4800	SARS-CoV-2 S-protein delivered intradermally via CELLECTRA^®^ EP system	ID	Inovio Pharmaceuticals	[77]
29		ZyCoV-D	First COVID-19 DNA vaccine (spike gene) induces immunity	ID	Cadila Healthcare	[78]
30		GX-19	Encodes SARS-CoV-2 S1 and S2, using vaccine vector pGX27	IM	Genexine	[79]
31		GX-19N	Induces broad T cell responses, potentially cross-reactive	IM	Genexine	[80]
32	RNA-based vaccine	BNT162b2	Encodes full-length spike (prefusion conformation), robust T cell responses	IM	Pfizer/BioNTech	[81]
33		BNT162b2 BA.1 bivalent booster	Bivalent mRNA encoding the original Wuhan-Hu-1 strain and BA.1	IM	Pfizer/BioNTech	[82]
34		BNT162b2 BA.4/5 bivalent booster	Bivalent mRNA encoding the original Wuhan-Hu-1 strain and BA.4/5	IM	Pfizer/BioNTech	[83]
35		BNT162b2 Monovalent XBB.1.5	XBB.1.5-specific changes in S spike protein, based on original BNT162b2	IM	Pfizer/BioNTech	[84]
36		Spikevax (mRNA-1273)	LNP-encapsulated mRNA vaccine encoding prefusion-stabilized SARS-CoV-2 S protein	IM	Moderna and NIAID VRC	[85]
37		mRNA-1273.214	Encoding ancestral Wuhan-Hu-1 and Omicron BA.1 spike mRNAs	IM	Moderna	[86]
38		mRNA-1273.222	Encoding ancestral SARS-CoV-2 and BA.4/5 spike proteins			[87]
39		mRNA-1273.815	Encoding ancestral SARS-CoV-2 and XBB.1.5 spike			
40		RQ3013	Pseudouridine-modified mRNAs in LNP encode S protein with B.1.1.7/B.1.351 mutations	IM	Shanghai Lanque Fudan University	[88]
41		RQ3033	Targets XBB.1.5, designed to prevent COVID-19 caused by XBB and EG.5	IM		[89]
42		CVnCoV	Sequence-engineered mRNA encoding SARS-CoV-2 S protein, protected by LNP delivery	IM	CureVac	[90]
43		CV2CoV	2nd vaccine with optimized non-coding regions, enhanced antigen expression	IM		[91]
44		SYS6006	Encodes S protein with S-2P, induces nAbs against WT, Delta, et al. in mice/NHPs	IM	CSPC Group	[92]
45		ARCT-154	Self-amplifying mRNA with modified S-protein	IM	Arcturus Therapeutics	[93]
46		VLPCOV-01	LNP-encapsulated RNA vaccine expressing membrane-anchored SARS-CoV-2 RBD	IM	Akahata W, et al.	[94]
47	Viral vector vaccine	Vaxzevria (AZD1222)	ChAdOx1 vector encoding full-length SARS-CoV-2 spike with S-2P mutation	IM	AstraZeneca-Oxford	[95]
48		Gam-COVID-Vac	RAd26 and rAd5 vectors carrying SARS-CoV-2 spike glycoprotein gene	IM	GRI	[96]
49		Ad5-nCoV	Ad5 vector encoding full-length SARS-CoV-2 spike with S-2P mutation	IM	CanSino Biologics	[97]
50		Ad26.COV2. S	Ad26 vector encodes pre-fusion stabilized full-length SARS-CoV-2 spike protein	IM	Johnson & Johnson	[98]
51		dNS1-RBD	Cold-adapted H1N1 NS1-deleted strain with inserted SARS-CoV-2 RBD	IN	Beijing Wantai	[99]
52		BBV154 (iNCOVACC)	ChAd36 vector encodes pre-fusion stabilized SARS-CoV-2 spike with S-2P	IN/IM	Bharat Biotech	[100]
53		AdCOVID	Intranasal Ad5 vectored vaccine encoding SARS-CoV-2 RBD	IN	Altimmune	[101]
54		CVXGA1	Recombinant PIV5 with SARS-CoV-2 spike, cytoplasmic tail replaced by PIV5 F	IN	University of Georgia	[102]
55		NDV-HXP-S	Modified spike with six prolines, swapped domains for NDV integration	IN	PATH	[103]
56		Patria	Live NDV vector vaccine expressing SARS-CoV-2 spike with S-2P mutation	IN	Avimex	[104]
57		MVA-SARS-2-S	MVA vector vaccine expressing full-length SARS-CoV-2 S protein	IM	DZIF	[105]
58		MVA-SARS-2-ST	With modified, stabilized SARS-CoV-2 S antigen and inactivated S1/S2 cleavage site	IM	DZIF	[106]
59		MV-014-212	RSV OE4 with SARS-CoV-2 S and RSV F tail	IM	Meissa Vaccines Inc	[107]
60		GBP510 (SKYCovione)	Targets RBD of S protein, uses AS03 adjuvant to boost reactogenicity	IM	SKB	[108]

This table summarizes key information on various SARS-CoV-2 vaccines, including vaccine platforms, product names, detailed compositions, immunization routes, and developing institutions. It encompasses multiple vaccine types, such as inactivated virus vaccines, live-attenuated vaccines, protein-based vaccines, VLP vaccines, DNA-based vaccines, RNA-based vaccines, and viral vector vaccines. The table offers comprehensive insights into the composition, technical features, and administration methods of each vaccine, serving as a valuable reference for understanding the development progress and characteristics of different SARS-CoV-2 vaccine platforms. IM, intramuscular injection; IN, intranasal injection; ID, intradermal injection.

Two prominent inactivated vaccines, BBIBP-CorV and CoronaVac, developed independently in China, have been approved by the World Health Organization (WHO). BBIBP-CorV uses β-propiolactone to inactivate the SARS-CoV-2 and has been shown to elicit strong neutralizing antibody responses against SARS-CoV-2 before the emergence of the Omicron variant [53]. Although the efficacy of BBIBP-CorV has been reduced by VOCs as a first-generation vaccine, combining it with other vaccine types may enhance its effectiveness. Similarly, CoronaVac, which uses chemically inactivated SARS-CoV-2 and aluminum hydroxide as an adjuvant, demonstrated improved neutralizing antibody titers against Omicron when combined with mRNA vaccines (BNT162b2 or mRNA-1273), highlighting the potential of hybrid vaccination approaches [109,110].

Inactivated vaccines contain non-replicating, killed viruses, making them generally safe; however, they may induce a weaker immune response compared to live-attenuated vaccines [111]. To achieve optimal protection, inactivated vaccines typically require at least two doses. Therefore, adjuvants are crucial for enhancing their immune response. One promising adjuvant is IMDG, a Toll-like receptor 7/8 agonist, which boosts adaptive immune responses by stimulating innate immune pathways [54]. For instance, Covaxin (BBV152), developed by Bharat Biotech and adjuvanted with Algel-IMDG, has demonstrated robust protective immunity, particularly against severe SARS-CoV-2 infections [54].

Unlike inactivated vaccines, live-attenuated vaccines enhance the immunogenicity of antigens by using weakened forms of specific viruses. A notable example is COVI-VAC, an intranasal live-attenuated vaccine designed to mimic natural SARS-CoV-2 infection and stimulate both mucosal and systemic immunity [60]. The development of COVI-VAC involved molecular recoding of a defined S protein domain and elimination of the Furin proteolytic cleavage site, resulting in an enhanced safety profile. COVI-VAC administered intranasally elicited robust neutralizing antibody titers comparable to those observed in SARS-CoV-2-infected Syrian golden hamsters. A temperature-adapted live-attenuated SARS-CoV-2 vaccine was also developed and tested in K18-hACE2 mice [112]. Administered as a single intranasal dose, it induced potent NAbs, cellular immunity, and mucosal IgA responses, which are crucial for preventing viral entry at mucosal surfaces.

### 3.2. Protein Subunit Vaccine

Recombinant protein vaccines are designed to produce specific antigenic proteins, such as viral S or capsid proteins, to stimulate an immune response. Subunit vaccines are typically produced by isolating viral antigens through in vitro protein expression. As this type of vaccine often exhibits relatively low immunogenicity, it requires the incorporation of adjuvants and multiple booster doses to achieve optimal immune responses. Common adjuvants include aluminum hydroxide, Matrix-M, and AS03. An example is GBP510 (SKYCovione), developed by SK Bioscience in South Korea, which targets the RBD and uses the AS03 adjuvant to enhance immune responses [108]. The combination of GBP510 with AS03 has been shown to increase reactogenicity compared to both the non-adjuvanted and placebo groups. Another recombinant protein vaccine, ZF2001, employs a dimeric RBD antigen expressed in Chinese hamster ovary (CHO) cells, with aluminum hydroxide as an adjuvant [113]. Administered in a three-dose regimen, ZF2001 has demonstrated strong immunogenicity, eliciting high levels of NAbs and robust T cell responses in clinical trials [114]. Similarly, MVC-COV1901, developed by Medigen Vaccine Biologics, also utilizes CHO cells and contains CpG 1018 and aluminum hydroxide as adjuvants [69]. CpG 1018, a toll-like receptor 9 (TLR9)-agonist oligodeoxynucleotide, enhanced immunogenicity and induced Th1-skewed responses in preclinical studies. A Phase 1 dose-escalation trial evaluated the safety and immunogenicity of three MVC-COV1901 doses administered twice (28-day interval) in healthy adults. Results demonstrated that 15 μg S-2P protein with CpG 1018/aluminum hydroxide elicited robust humoral immunity [115]. Alternatively, Corbevax, produced using the Pichia pastoris yeast platform and adjuvanted with aluminum hydroxide and CpG 1018, is widely used in India to enhance immunogenicity [116]. Additionally, the baculovirus expression vector system (BEVS) is another efficient method for producing recombinant proteins in insect cells, which closely mimic the post-translational modifications found in mammalian cells. An example of a vaccine produced using BEVS is NVX-CoV2373, a protein subunit vaccine that uses an Sf9 insect cell-expressed, prefusion-stabilized S protein combined with the Matrix-M adjuvant and has been authorized by the WHO for emergency use [117].

While global vaccinations and prior SARS-CoV-2 infections elevate antibody levels, Omicron variants continue spreading despite high vaccination rates. Sera from recipients of WT/Delta vaccines show reduced neutralization against JN.1 and its sublineages, potentially insufficient for infection protection, highlighting the need for effective vaccines. Given the significant spike sequence divergence between JN.1 and XBB.1.5, concerns exist that XBB.1.5-based vaccines may inadequately protect against emerging JN.1 lineages. However, updated XBB protein vaccines, such as WSK-V102C (XBB.1.5+BA.5+Delta) and WSK-V102D (XBB) boosters, demonstrate promising efficacy against broad Omicron variants including BA.2.86 and JN.1. Notably, significantly decreased neutralization against JN.1.13, KP.2, and KP.3 compared to JN.1 in boosted adults. These findings underscore the enhanced neutralization resistance of JN.1 subvariants and emphasize the critical need for boosters targeting currently circulating variants [118].

Notably, while adjuvants play a significant role in enhancing vaccine effectiveness, the AKS-452 protein subunit vaccine utilizes the Ambifect™ Fc-fusion protein platform (SP/RBD-Fc) to boost neutralizing IgG titers and stimulate a mixed Th1/Th2 immune response against the SARS-CoV-2 RBD [119]. The Fc region enhances immunogenicity by promoting antigen uptake by antigen-presenting cells (APCs) through FcγR and prolonging exposure via the neonatal Fc receptor (FcRn), ensuring better circulation of the antigen [120]. Additionally, the inclusion of the water-in-oil adjuvant Montanide™ ISA 720 further amplifies its immunogenicity [121].

### 3.3. Virus-like Particle Vaccine

To induce a stronger and broader immune response, nanoparticle-based vaccines offer a promising strategy by presenting multiple dominant antigenic epitopes, thereby enhancing vaccine potency. VLP vaccines, among the earliest protein nanoparticle vaccines, have been widely used as demonstration platforms. These vaccines are based on the assembly of viral proteins that mimic the structure of the virus but lack its genetic material, making them safer for human use [122]. Plant-derived VLPs may have a significant safety advantage, as the risk of contamination with human pathogens is extremely low [121,123]. Early attempts at VLP production in plants faced several drawbacks, including low yields [124]; however, recent advancements in plant expression systems have enabled the production of large quantities of recombinant protein with post-translational modifications, allowing for VLP assembly. Covifenz, a VLP-based vaccine developed by Medicago using plant-derived VLPs, demonstrated 75.3% efficacy against COVID-19 caused by the Delta variant and 88.6% efficacy against the Gamma variant [125]. This development offers a low-cost solution with a low risk of introducing adventitious human pathogens [123,126].

Another type of nanoparticle, self-assembled proteins, contains self-assembling motifs that enable soluble proteins to spontaneously assemble into protein nanoparticles. Joyce et al. developed an in vivo SARS-CoV-2 self-assembled nanoparticle vaccine using ferritin as a platform, which demonstrated broad neutralizing antibody responses against respiratory infection and disease in nonhuman primates [127]. One example, bacterial ferritin nanoparticles display antigens to reconstitute trimeric class I glycoproteins and enhance immunogenicity for weak targets. Incorporating the HR subdomain leverages its high conservation across coronaviruses, especially betacoronaviruses. HR-displaying vaccines elicit broadly cross-reactive neutralizing antibodies, demonstrating pan-coronavirus potential. RBD-HR nanoparticles specifically boost germinal center responses, increasing Tfh/B cell frequencies and RBD-specific IgG1/IgG2b memory B cells [128]. Additionally, in vitro self-assembled nanoparticles have also shown enhanced antibody responses against multiple antigens, as demonstrated with pre-fusion RSV and HIV-1 envelope proteins [129,130]. Furthermore, Walls et al. multivalently displayed 60 SARS-CoV-2 RBD molecules on the exterior surface of the two-component protein nanoparticle I53-50, eliciting highly immunogenic responses and significantly higher neutralizing antibody titers [131].

Taken together, recombinant protein vaccines, including those incorporating nanoparticles, have demonstrated both safety and high efficacy, playing a pivotal role in addressing the global challenge posed by SARS-CoV-2. As the virus continues to evolve, ongoing advancements in antigen design, nanoparticle technology, and adjuvant formulations will be essential to ensuring the long-term effectiveness of these vaccines and preparedness for future pandemics.

### 3.4. Nucleic Acid Vaccines

Nucleic acid vaccines represent one of the most innovative and rapidly developed platforms for SARS-CoV-2 vaccination. Unlike traditional vaccines, which use protein antigens, nucleic acid vaccines deliver genetic material—either mRNA or DNA—that encodes the SARS-CoV-2 S protein into host cells [132]. These cells then use the genetic instructions to synthesize the S protein, which is subsequently recognized by the immune system, stimulating both humoral and cellular immune responses.

mRNA vaccines targeting the SARS-CoV-2 S1 protein have demonstrated exceptional efficacy, setting a new benchmark for COVID-19 vaccine development. The most well-known examples of nucleic acid vaccines for COVID-19 are Pfizer-BioNTech’s BNT162b2 and Moderna’s mRNA-1273. Both vaccines utilize lipid nanoparticles to deliver mRNA encoding the S protein. BNT162b2 was the first mRNA vaccine to receive emergency use authorization globally [133]. In Phase III trials, BNT162b2 demonstrated 95% efficacy against symptomatic COVID-19 [81]. While the BNT162b2 vaccine maintained broad-spectrum efficacy against dominant SARS-CoV-2 variants during the early phases of the pandemic [134], the emergence of antigenically divergent Omicron sublineages exhibiting enhanced immune evasion mechanisms has progressively eroded its protective capacity against mild symptomatic infections [135]. Booster doses are crucial to enhance immunity, particularly against newer strains. Two doses of BNT162b2 demonstrated limited efficacy against all BA.4/5-related outcomes, including hospitalization. While booster doses (third or fourth) conferred temporary protection, their durability appears time-dependent: protection against mild outcomes waned after ~3 months, whereas protection against severe outcomes persisted for ~6 months [136]. In addition, the bivalent BNT162b2 booster significantly increases neutralizing antibody titers against Omicron variants [137]. Clinical evidence supports the favorable benefit–risk ratio of the BNT162b2-Omi.BA.4/BA.5 booster, as evidenced by 1-month immunogenicity and safety outcomes in triple-primed recipients of the original BNT162b2 series [138]. Additionally, combining BNT162b2 with other vaccine platforms, such as inactivated vaccine CoronaVac, has been shown to enhance the immune response, providing broader protection and improving efficacy against diverse SARS-CoV-2 variants [139]. Similarly to BNT162b2, Moderna’s mRNA-1273 shares a comparable mechanism of action. Notably, bivalent vaccines co-formulated with either the SARS-CoV-2 prototype and BA.1 (mRNA-1273.214) or BA.4/5 (mRNA-1273.222) induced superior heterotypic immunity in mouse models. These formulations targeting Omicron strains provide substantial protection against both symptomatic infections and severe disease [140]. The 2023/24 updated vaccine formulation targeted the XBB.1.5 sublineage. Vaccine effectiveness (VE) against COVID-19 hospitalization was 50–70% during the first three months post-vaccination, aligning with previous estimates. Despite waning efficacy and emerging immune-evading variants like JN.1, XBB.1.5 mRNA vaccination maintained significant, durable protection against COVID-19 hospitalizations [141]. More recently, CureVac’s CVnCoV developed a second-generation vaccine (CV2CoV), which optimizes untranslated regions (UTRs) and employs modified mRNA technology to enhance protein expression and stability [91]. The improved translation efficiency and strong immune activation of CV2CoV highlight its potential to offer broad protection against emerging variants, addressing the limitations of earlier mRNA vaccine formulations. Despite the weakened neutralizing antibody response, these vaccines continue to prevent severe diseases caused by most of these VOCs.

Another type of nucleic acid-based vaccine is DNA vaccines, which have been explored for SARS-CoV-2. ZyCoV-D, the first DNA vaccine to receive emergency approval for COVID-19, induces both humoral and cellular immunity, making it a promising option for boosting immunity, particularly in regions where mRNA vaccine distribution faces logistical challenges [78]. Similarly, INO-4800, another DNA-based vaccine, shows strong potential in enhancing T cell memory [77]. Unlike antibody responses, T cell memory offers more durable protection against reinfection, highlighting its crucial role in sustaining long-term immunity [142].

Nucleic acid vaccines offer several advantages, including rapid development, scalability, and the ability to be quickly adapted in response to emerging variants. During the pandemic, mRNA vaccines have gained global prominence. Their rapid development and widespread deployment across numerous countries demonstrate significant potential. This successful implementation establishes mRNA technology as a validated platform for future pandemic responses. However, mRNA vaccines face significant storage challenges, although the development of thermostable formulations or drying processes such as lyophilization, spray drying, and spray-freeze drying has partially mitigated this issue [143,144]. In contrast, DNA vaccines offer greater stability and are easier to store, making them more suitable for distribution in resource-limited settings. Nonetheless, the need for electroporation devices to optimize DNA vaccine delivery remains a limitation, potentially restricting their widespread use compared to mRNA vaccines [145]. Future research should focus on enhancing the storage stability, duration of immune response, and breadth of protection against new variants for nucleic acid vaccines. Notably, self-amplifying RNA and DNA vaccines, which enhance antigen production within the host, are emerging as promising candidates [146]. Additionally, hybrid vaccine strategies that combine mRNA vaccines with multimodule DNA nanostructure-assembled compartments or viral vector platforms are being explored to provide a more robust and durable immune response [147,148].

### 3.5. Vector Vaccines

Vector vaccines employ a different virus, such as an adenovirus, as a carrier to deliver genetic material encoding the viral protein. These vaccines have been instrumental in the global effort to control the COVID-19 pandemic, as they induce both humoral and cellular immune responses, potentially leading to longer-lasting immunity.

Adenoviral-based vaccines, including ChAdOx1-S, Ad26.COV2.S, Sputnik V, and Convidecia have demonstrated substantial efficacy in preventing severe disease and hospitalization. The ChAdOx1-S vaccine, developed by the University of Oxford and AstraZeneca, utilizes a chimpanzee adenovirus vector to deliver the SARS-CoV-2 S protein gene [149]. Booster doses of ChAdOx1-S have been shown to enhance immunity, particularly against Omicron subvariants [150]. Furthermore, heterologous boosting with ChAdOx1-S and BNT162b2 demonstrates superior neutralizing activity levels at three months post-vaccination, compared to homologous mRNA vaccination [151]. However, by the sixth month, neutralizing levels decline in both regimens, as evidenced by the plaque reduction assay against the Delta variant. Similarly, Ad26.COV2.S vaccine, developed by Johnson & Johnson, employs an adenovirus type 26 (Ad26) vector and demonstrated 52.0% and 64.0% efficacy in preventing moderate to severe COVID-19 in initial studies [98]. Although its effectiveness against Omicron infection is reduced, it continues to offer strong protection against hospitalization and severe disease [152]. Likewise, the Sputnik V vaccine (Gam-COVID-Vac), developed by the Gamaleya Research Institute in Russia, utilizes a heterologous adenoviral vector regimen (Ad26 and Ad5) in a two-dose schedule, inducing robust immune responses characterized by high levels of NAbs and strong T cell activation [153]. Convidecia, a single-dose adenoviral vector vaccine, has also elicited strong immune responses, including high levels of NAbs and robust cellular immunity. This feature is particularly relevant for elderly individuals, as heterologous boosting with Convidecia following a two-dose priming regimen enhances long-term protection [154]. In contrast, low neutralizing antibody responses against Omicron variants were observed in individuals who received two or three doses of the widely used inactivated vaccine, CoronaVac, demonstrating clear immune evasion [155]. Notably, the breadth of the enhanced immune responses following heterologous immunization with Convidecia was greater than that observed with the homologous regimen in individuals aged 60 years or older [154].

The effectiveness of vector-based vaccines varies based on the type of viral vector and formulation approach. Adenoviral vectors, such as ChAdOx1-S, Ad26.COV2.S and Sputnik V have been extensively studied and widely deployed, eliciting strong immune responses that stimulate both the innate and adaptive immune systems, offering broad protection against SARS-CoV-2. The ChAdOx1-S vaccine demonstrated 72% efficacy against symptomatic SARS-CoV-2 infection. Vaccine effectiveness tended to increase with longer intervals between doses. However, a key challenge associated with these vaccines is the presence of pre-existing immunity to the viral vectors, which may reduce the efficacy of booster doses [156,157]. Although ChAdOx1-S was extensively utilized early in the pandemic owing to rapid production capacity, subsequent studies identified associations with rare but severe Thrombosis with Thrombocytopenia Syndrome (TTS), notably cerebral venous sinus thrombosis (CVST) [158]. These findings prompted risk–benefit reassessments indicating that potential risks may outweigh benefits in younger adult populations. Although Ad26 has a low global seroprevalence, repeated exposure to Ad26-based vaccines may lead to the development of anti-vector immunity, potentially diminishing antigen-specific immune responses [159]. To mitigate this issue, the development of non-adenoviral vector vaccines, such as ZyCoV-D, represents a promising advancement. ZyCoV-D employs a needle-free delivery system, which may be particularly beneficial for populations hesitant about injections [78]. Furthermore, ZyCoV-D has been shown to elicit a robust secondary anamnestic immune response upon re-exposure, mediated by balanced activation of memory B cells and helper T cells, thereby offering strong potential for long-term immunity [78].

In short, vector-based vaccines, particularly those using adenoviral platforms, have proven highly effective in combating COVID-19 and remain crucial in the global vaccination effort. Despite challenges such as pre-existing immunity to adenoviruses and reduced efficacy against certain variants, these vaccines continue to play a key tool in achieving worldwide vaccination targets. Due to safety issues and storage challenges, adenovirus vaccines may now be used mainly for first vaccinations in areas with limited resources, rather than as a primary choice for booster programs. Ongoing research into non-adenoviral vector vaccines and alternative delivery systems will be essential for improving the flexibility and durability of vaccination strategies against COVID-19 and future pandemics.

## 4. Neutralizing Antibodies Against SARS-CoV-2

### 4.1. RBD Targeting

NAbs play a crucial role in controlling viral infections. Based on their targeted epitopes, SARS-CoV-2-specific NAbs can be classified into three distinct categories, as summarized in Table 2. To highlight antibodies with broad-spectrum efficacy, Table 3 presents a selection of antibodies exhibiting pan-sarbecovirus neutralizing activity, along with an overview of those currently identified. The RBD of the SARS-CoV-2 S protein is a key target for monoclonal antibody (mAb) development due to its pivotal role in viral entry. Consequently, antibodies targeting this region can effectively inhibit this essential step, exhibiting strong antiviral activity [160]. This section explores various RBD-targeting antibodies, their neutralization mechanisms, and the challenges posed by viral mutations that impact their efficacy.

As mentioned above, Barnes et al. classified RBD-targeting antibodies into four categories to elucidate their neutralization mechanisms and potency differences [29]. Class I antibodies, such as CB6, bind the RBD in the “up” conformation and block ACE2 interaction by overlapping with the receptor-binding motif (RBM) [37]. However, its neutralization potency is significantly reduced against the Omicron BA.1 variant, which carries K417N, E484A, and N501Y mutations that facilitate immune escape [199]. Class II antibodies, including Bamlanivimab (LY-CoV555) developed by Eli Lilly, bind to the RBD in both the “up” and “down” conformations, preventing ACE2 interaction [200]. Although Bamlanivimab effectively neutralizes wild-type SARS-CoV-2, its efficacy is reduced against variants such as Delta and Omicron due to immune-evasive RBD mutations [201]. In contrast, class III and class IV antibodies exhibit broader antiviral neutralization compared to class I and II antibodies, as they bind to more hidden epitopes. For example, Sotrovimab (S309), a class III mAb, binds outside the ACE2-binding region, enabling recognition of the RBD in both conformations [164]. Sotrovimab has shown consistent efficacy against SARS-CoV-2 variants prior to BA.1 [202]; however, its efficacy was impacted by the emergence of the BA.2 variant, leading to the withdrawal of its emergency use authorization in the US (FDA update 5 Aril 2022). CR3022, a class IV mAb originally isolated from a SARS-CoV-1 convalescent patient, targets a conserved RBD epitope that does not directly engage ACE2 but disrupts structural rearrangements critical for viral fusion [203]. Although CR3022 exhibits cross-reactive binding among sarbecoviruses, its neutralization potency against SARS-CoV-2 is limited, as it binds only when at least two RBDs adopt the “up” conformation.

SARS-CoV-2 variants continue to accumulate mutations in the RBD, reducing antibody binding affinity and facilitating immune escape. Recent studies highlight the ongoing viral evolution, with emerging variants such as BA.2.86 and JN.1 carrying R346S/T, F456L/V, and A475V/S mutations, while the KP.3 variant harbors the unprecedented Q493E mutation [204,205]. The FLiRT variant harbors R346T and F456L mutations in the S1 subunit and V1104L in S2. Mechanistically, R346T compensates for ACE2 affinity loss caused by L455S/F456L mutations: F456L disrupts hydrophobic interactions in the RBD, weakening ACE2 binding, while R346T restores stability through conformational reinforcement in the RBM. The KP.2-characteristic V1104L mutation stabilizes the spike trimer via hydrophobic core packing, potentially impeding the prefusion-to-postfusion transition and thus reducing infectivity [206]. Many of these mutations reside near the RBM, posing substantial challenges to the efficacy of RBD-targeting antibodies and vaccines. Interestingly, while viral mutations drive the emergence of new variants, they may also enhance immunogenicity. The R452-specific antibody ConD-852, isolated from a Delta-infected donor, indicates that the L452R mutation—shared by various emerging variants and vaccine strains—can potentiate neutralizing antibody responses [207].

### 4.2. NTD Targeting

While the RBD has garnered significant attention as a target for NAbs, the NTD also plays a crucial role in viral neutralization. As part of the S1 subunit, the NTD is located on the viral surface, making it an accessible target for the immune system [208]. Several immunodominant epitopes within the NTD have been recognized as targets of NAbs. One of the primary mechanisms of NTD-specific NAbs is the inhibition of virus–host interactions. By binding to the NTD, these antibodies can prevent viral entry into the host cell by disrupting virus-receptor interactions. For example, BLN14 binds to the NTD, thereby interfering with the interaction between the virus and the L-SIGN receptor, as well as other associated elements [209]. Furthermore, some NTD-targeting antibodies induce conformational changes in the S protein, preventing the structural rearrangements required for membrane fusion and ultimately inhibiting viral replication [174,210].

Most NTD-targeting NAbs recognize an antigenic epitope known as the “NTD supersite” [33]. For instance, antibodies such as 4–8, 5–24, and S2M28 specifically target this supersite, which is crucial for neutralizing early emerging VOCs [193,211]. However, many circulating SARS-CoV-2 variants harbor mutations within this supersite, potentially reducing the neutralization potency of these antibodies. For example, the V213G mutation, found in multiple Omicron subvariants, has been associated with significant immune evasion against NTD-targeting antibodies [212]. Specifically, deletions of amino acid residues 144, and 242–244, along with the mutation at position 246, have been shown to severely impair the neutralizing ability of antibodies such as 4–8 and 5–24, rendering them almost ineffective against the Beta variant [213]. In contrast, antibodies targeting non-supersite regions of the NTD, such as 5–7, exhibit broader neutralization against various SARS-CoV-2 strains compared to supersite-directed antibodies [214]. However, they still experience some reduction in neutralizing activity, with an approximately 80-fold reduction against the BA.1 variant [213]. The continuous accumulation of mutations in the NTD has led to significant immune escape. For example, the XEC variant carries two additional mutations, T22N and F59S, in the NTD, which are absent in the KP.3 variant [215]. Compared to its parental strain KP.3, the XEC variant shows increased infectivity, largely attributed to the F59S mutation, while the T22N mutation has a negligible effect [215]. Notably, the T22N mutation introduces an N-linked glycosylation site, which may obscure antibody recognition and enhance immune evasion. Structurally, the T22N mutation generates an N-linked glycosylation, adding a surface-exposed glycan that sterically obstructs a key NTD antibody epitope. This modification impairs neutralizing function of epitope-specific antibodies like through glycan interference, thereby facilitating viral immune escape by diminishing NTD-targeted neutralization and promoting evasion of pre-existing immunity [216]. Furthermore, KP.3.1.1, currently the most prevalent subvariant globally, lacks the S31 deletion observed in KP.3. This deletion has been independently detected in several distinct JN.1 sublineages, including KP.2.3, LB.1, KP.3.1.1, and LF.2. The convergent acquisition of the S31 deletion has been associated with both enhanced immune evasion and an increased relative effective reproduction number compared to JN.1 subvariants lacking this deletion [217,218]. Interestingly, the S31 deletion in KP.3.1.1 introduces a potential N-linked glycosylation site (PNGS, NxS/T motif) at N30 (NFT), which may alter the local S conformation and reduce the efficacy of pre-existing NAbs [219]. This change may promote a more “down” conformation of the RBD, further hindering antibody recognition.

Beyond mutations in the NTD supersite, additional glycosylation sites introduced by asparagine residues can further shield epitopes from antibody binding. For example, Zhang et al. defined a non-supersite NTD-targeting neutralizing antibody, 3711, which recognizes a “silent face” epitope that is partially obscured by glycan structures [176]. This antibody effectively protects mice from infection by wild-type SARS-CoV-2 and other variants. These findings highlight the potential of mAbs targeting non-supersite regions of the NTD as a promising strategy for mitigating immune evasion and improving COVID-19 therapeutic interventions.

### 4.3. S2 Domain Targeting

As mentioned above, the S2 domain plays a critical role in mediating the fusion of viral and host cell membranes, which is a key step in viral entry. Given its pivotal function in the viral life cycle, the S2 domain has become a promising target for therapeutic antibodies that block infection at an early stage [220]. Compared to the RBD, the S2 domain exhibits a higher degree of conservation across SARS-CoV-2 variants, making it an attractive target for broadly NAbs. This enhanced conservation suggests that S2-targeting therapies could offer more lasting and comprehensive protection against emerging variants.

Several recently identified NAbs targeting the FP or SH regions within the S2 subunit have demonstrated broad cross-neutralizing activity against multiple coronaviruses. For instance, VN01H1, VP12E7, and C77G12, isolated from SARS-CoV-2 convalescent and vaccinated individuals, target a core epitope in the FP region [36]. VN01H1 and VP12E7 block the entry of both α- and β-coronaviruses, while C77G12 exhibits stronger neutralization activity against β-coronaviruses. All three antibodies inhibit SARS-CoV-2 S protein-mediated cell–cell fusion. Another antibody, 76E1, targets an epitope encompassing both the FP and the S2′ cleavage site, conferring cross-neutralizing activity against α- and β-coronaviruses [221]. Additionally, 76E1 demonstrates cross-binding activity to peptides containing epitopes from γ- and δ-coronaviruses, broadening its therapeutic potential. Notably, FP-targeting NAbs enhance binding in an ACE2-dependent manner, in contrast to SH- or RBD-targeted NAbs, suggesting that the FP epitope becomes exposed following receptor-binding-induced conformational changes [222].

The SH region also represents a viable target within the S2 subunit. For example, the mAb S2P6 binds to the viral S protein and is thought to disrupt stem-helical bundles, thereby preventing the conformational changes required for membrane fusion and subsequent viral entry [197]. S2P6 exhibits broad neutralization against β-coronaviruses, including sarbecoviruses (SARS-CoV-1 and SARS-CoV-2), merbecovirus (MERS-CoV), and embecovirus (HCoV-HKU1 and HCoV-OC43), with IC_50_ values ranging from 1.3 to 17.1 μg/mL. Another SH-targeting antibody, CC40.8, isolated from a patient recovering from COVID-19, also demonstrates broad neutralizing activity against β-coronaviruses [179]. These antibodies neutralize coronaviruses by destabilizing the pre-fusion S conformation, thereby effectively preventing membrane fusion.

Unlike RBD-targeting antibodies, which prevent viral entry by blocking ACE2 binding, S2-targeting antibodies primarily inhibit the conformational changes necessary for membrane fusion [223]. However, certain neutralizing epitopes on the S2 subunit may be shielded by N-glycans or other structural elements of the trimeric S protein, potentially limiting immune recognition. Therefore, when developing broad-spectrum SARS-CoV-2 antibodies or vaccines, it is crucial to account for glycosylation and cryptic epitopes that could hinder effective immune targeting.

## 5. Potential Strategies to Optimize COVID-19 Vaccines

### 5.1. Broadly NAbs Drive the Development of Broad-Spectrum Vaccines

Since the early 21st century, numerous viral outbreaks have posed significant challenges to global public health and the socio-economic landscape. Concern over coronaviruses in human health intensified following the outbreaks of SARS-CoV-1 in 2002–2003 and MERS-CoV in 2012. This concern was further heightened by the emergence of SARS-CoV-2 in December 2019, which posed a major global health threat. SARS-CoV, MERS-CoV, and SARS-CoV-2 all belong to the β-coronavirus genus, underscoring the urgent need for broad-spectrum NAbs to combat both current and future coronavirus threats. To this end, we review major vaccine design strategies aimed at guiding the development of effective interventions against SARS-CoV-2, other sarbecoviruses, and β-coronaviruses more broadly (Figure 3).

As mentioned above, RBD-targeting antibodies exhibit high efficacy against SARS-CoV-2, the substantial variability of the RBD among SARS-CoV-2 variants and across different coronavirus species can limit their effectiveness. To this end, multispecific antibodies combine the binding domains of multiple mAbs within a single molecule, reducing viral escape risk by targeting multiple epitopes simultaneously. Our study demonstrates that Tri-1 and Tri-2 represent promising platforms for developing multivalent therapeutics, leveraging cross-neutralizing antibodies to combat circulating SARS-CoV-2 variants and phylogenetically related sarbecoviruses (Pangolin-GD, RaTG13, WIV1, and SHC014) [224]. On the other hand, the S2 region of the coronavirus spike represents a relatively conserved alternative target that contains neutralizing epitopes, making it promising for developing vaccines effective against SARS-CoV-2 VOCs and potentially pan-betacoronaviruses. The more conserved S2 subunit represents a promising target for broad-spectrum antibody development against a broader range of coronaviruses, including both α- and β-coronaviruses, albeit with somewhat reduced in vitro neutralization potency [222]. Notably, FP-targeted NAbs, such as COV44-62 and COV44-79, exhibit broad-spectrum neutralizing potency against α-CoVs, including HCoV-NL63 and HCoV-229E, as well as β-CoVs, including SARS-CoV-2, SARS-CoV-1 and HCoV-OC43 [38]. Additionally, the shark-derived nanobody 79C11, targets a conserved HR1 epitope in the S2 region to block membrane fusion and exhibits pan-sarbecovirus neutralizing activity, including against emerging SARS-CoV-2 variants (XBB.1.5, JN.1, and KP.2), SARS-CoV-1, and pangolin CoV [225]. Thus, identifying and optimizing NAbs that target conserved epitopes within the S protein could offer a promising approach for developing effective broad-spectrum vaccines.

It is worth noting that the pursuit of broad-spectrum COVID-19 vaccines faces significant scientific and practical hurdles, mirroring the decades-long struggle to develop universal influenza vaccines. Conserved epitopes, such as the S2 stem helix, typically induce weaker neutralizing responses than strain-specific RBD epitopes. Current limitations involve significant antigenic variation across coronaviruses and insufficient evidence for cross-protective immunity. Persistent immune escape and limited cross-reactivity render broad-spectrum candidates unlikely to eliminate booster needs, mirroring the irreplaceable role of seasonal influenza vaccines. Future vaccine development efforts need to prioritize epitope-targeted strategies validated through rigorous preclinical studies prior to clinical trial initiation. This strategy may contribute to more durable protection against both current and emerging coronaviruses.

### 5.2. Modification of Vaccine Antigen Composition

SARS-CoV-2 continues to circulate and evolve, presenting ongoing challenges to the effectiveness of current vaccines. A study demonstrated that individuals vaccinated with Ad26.COV2.S (single dose), Sputnik V (two doses), or BBIBP-CorV (two doses) exhibited minimal neutralizing activity against Omicron variants [226]. Similarly, vaccines such as mRNA-1273, BNT162b2, and AZD1222 maintained detectable neutralization against the wild-type SARS-CoV-2; however, their effectiveness against the Omicron variant was significantly reduced by 39-fold, 37-fold, and 21-fold, respectively [24]. These findings underscore the limited cross-protection and durability of existing vaccines, particularly against Omicron sublineages. Moreover, there is growing interest in developing hybrid or heterologous vaccine formulations, as adjusting the composition of COVID-19 vaccine antigens is crucial for enhancing efficacy. In May 2023, WHO recommended the use of a monovalent XBB.1-descendent lineage, such as XBB.1.5, as a vaccine antigen. As of April 2024, with the emergence of the JN.1 and its descendants, the WHO currently recommends a monovalent JN.1 lineage-based vaccine for future immunization efforts.

Additionally, insights can be drawn from the evolutionary trajectory of the influenza virus A (H5Nx). Unlike the relatively linear evolution of A (H3N2) and A (H1N1), A (H5Nx) exhibits multi-directional evolution with a high degree of genetic variation [227]. This pattern is comparable to that of SARS-CoV-2, which has a broad host range and a geographically dispersed distribution [228]. These factors collectively reduce the impact of immune selection. Therefore, vaccine strain selection should not be limited to the most recently emerged variants but should incorporate a broader range of virological and immunological considerations. This complexity presents significant challenges in selecting vaccine seed strains, as relying on only one or two virus strains may be insufficient to provide comprehensive protection. Multivalent or broad-spectrum vaccines offer a promising strategy for conferring immunity against both existing and emerging SARS-CoV-2 variants. For example, NVSI-06-08 (Sinopharm) incorporates a mutation-integrated trimeric RBD as its antigen, integrating key mutations from three heterologous SARS-CoV-2 variants (Wild-type, Beta, and Kappa) into a single protein [229]. Pre-clinical studies have demonstrated that NVSI-06-08 induces a broader immune response against various SARS-CoV-2 variants, suggesting its potential as a more versatile vaccine candidate. A similar approach has been employed in influenza vaccine research, where the chimeric hemagglutinin (cHA) vaccine strategy effectively induces a broad spectrum of protective antibodies and cellular immune responses against multiple influenza virus subtypes [230].

A mosaic nanoparticle approach has also demonstrated promise in eliciting broadly protective antibody responses. Mosaic-8b, a potential pan-sarbecovirus vaccine, features RBDs from SARS-CoV-2 and seven animal sarbecoviruses covalently attached to a 60-mer protein nanoparticle. Notably, mosaic-8-immunized animals exhibited greater cross-reactivity against sarbecoviruses compared to those immunized with homotypic SARS-CoV-2 vaccines [231]. Similarly, 3Ro-NC integrates structural components from multiple RBDs, including one Delta RBD and two Omicron RBDs. Furthermore, intranasal immunization with 3Ro-NC, in combination with the mucosal adjuvant KFD, effectively enhances neutralizing antibody specificity and elicits coordinated mucosal IgA responses against Omicron [232].

### 5.3. Advancing Long-Lasting Coronavirus Vaccines

Unlike SARS-CoV-2 natural infection, neutralizing antibody responses induced by standard vaccination typically decline to baseline levels within six months. The emergence of waning immunity and breakthrough infections underscores the urgent need for improved vaccines with enhanced immunogenicity and durability [233]. Booster doses have been shown to significantly elevate antibody levels, with heterologous or sequential vaccination regimens demonstrating even more pronounced enhancements. A study by Zuo et al. found that an mRNA booster following two doses of inactivated vaccines significantly boosts neutralizing antibody levels and memory cell responses against SARS-CoV-2 and its variants [234]. This heterologous inactivated/mRNA vaccination strategy holds great potential as an effective approach for enhancing immunity. In addition to antibody responses, booster doses also stimulate pre-existing T follicular helper (TFH) cells, which promote the differentiation of memory B cells into antibody-secreting cells [234]. Targeting TH1-like TFH differentiation through cytokines or adjuvants could provide valuable strategies for developing long-lasting coronavirus vaccines by sustaining durable immunity [235].

On the other hand, self-amplifying RNA vaccines represent another promising approach for developing long-lasting coronavirus vaccines. Based on mRNA technology, these vaccines incorporate engineered elements that enable RNA replication within host cells, enhancing the immune response while reducing the required dose [236]. This innovation makes self-amplifying RNA vaccines an attractive option for creating long-acting COVID-19 vaccines. Notably, the first self-amplifying RNA COVID-19 vaccine, ARCT-154, has been approved by regulators in Japan [237]. With a lower dose, it demonstrates safety and efficacy comparable to the mRNA vaccine BNT162b2 [238].

### 5.4. Establishing Mucosal Immunization Protection

Intramuscular injection remains the primary method of vaccine administration; however, it often fails to elicit robust mucosal immune responses in the upper respiratory tract, the first line of defense against SARS-CoV-2 infection. Secretory immunoglobulin A (sIgA) plays a crucial role in respiratory mucosal immunity [239]. During early infection or vaccination, plasma cells in mucosal tissues initially secrete low-affinity IgA or IgG, followed by the production of high-affinity IgA as antibody maturation progresses [240]. Over time, high-affinity dimeric IgA (dIgA) binds to the polymeric immunoglobulin receptor (pIgR) on epithelial cells via endocytosis, facilitating its transport to the respiratory mucosal surface, where it forms sIgA [241]. This process enables sIgA to block viral entry into the respiratory tract, providing broad-spectrum protection, including against Omicron variants.

Recent COVID-19 vaccine development has increasingly focused on enhancing mucosal immunity. Adenovirus vector-based vaccines administered nasally can overcome the limitations of intramuscular vaccination by directly targeting mucosal immunity. For example, Ad5-nCoV-IH vaccine, which employs an Ad5 delivery vector to express the full-length S protein of wild-type SARS-CoV-2, is delivered via aerosol inhalation [242]. Approved as a booster for emergency use in China in September 2022, this vaccine has demonstrated enhanced T cell responses and a favorable safety profile compared to intramuscular injections. A recent study conducted a head-to-head comparison of the effectiveness of booster doses administered via aerosolized inhalation (IH) and intramuscular (IM) injection of Ad5-nCoV. The results showed that IH Ad5-nCoV booster elicited a stronger immune response, though it did not provide significantly greater protection against SARS-CoV-2 (52.3%) compared to the IM route (37.2%), despite being administered at a dose four-fifths lower [243]. Similarly, iNCOVACC (BBV154), an intranasal vaccine encoding a prefusion-stabilized SARS-CoV-2 S protein, received emergency use authorization in India. Clinical trials indicated that individuals who received two intranasal doses of iNCOVACC, 28 days apart, exhibited higher sIgA levels than those who received the intramuscular Covaxin vaccine [244].

Beyond intranasal delivery, oral administration has also been explored as a strategy to induce mucosal immunity. MigVax-101, an oral multi-antigen SARS-CoV-2 vaccine, comprises the RBD of the viral S protein, two domains of the N protein, and the heat-labile enterotoxin B (LTB) as a mucosal adjuvant. Both a three-dose vaccination schedule and a heterologous subcutaneous prime followed by an oral booster successfully induced humoral, mucosal, and cell-mediated immune responses in preclinical studies [245].

Meanwhile, mucosal vaccines face persistent challenges due to the suboptimal immunogenicity of mucosally administered antigens, necessitating robust adjuvants or delivery systems to enhance adaptive immunity. KFD, a mucosal adjuvant, activates the TLR5 pathway in nasal epithelial cells, thereby promoting local and systemic mucosal IgA responses [246]. In a recent study, intranasal immunization with 3Ro-NC combined with KFD (3Ro-NC+KFDi.n) induced strong mucosal IgA responses and enhanced neutralizing antibody specificity against Omicron variants [232]. The careful selection of immune adjuvants tailored to distinct immunization modalities remains a critical determinant for successful vaccine development. Therefore, establishing mucosal immunization is critical for preventing SARS-CoV-2 infection and transmission.

### 5.5. Development of Combination Vaccines for Respiratory Infectious Diseases

Beyond COVID-19, respiratory pathogens such as influenza, respiratory syncytial virus (RSV), and human metapneumovirus (HMPV) remain significant public health threats, particularly in vulnerable populations, as they can cause severe pneumonia and exacerbate chronic cardiopulmonary conditions [247]. Consequently, the development of combination vaccines targeting COVID-19 and other respiratory viruses has become a critical focus in vaccine research. Co-infection with influenza and COVID-19 has caused severe disease during recent cocirculation periods. The simultaneous circulation of these viruses increases the risks of vaccination errors, as individuals needing COVID-19 vaccination may have recently received influenza vaccines, or vice versa. Furthermore, separate administrations of both vaccines are required to prevent dual infection, increasing healthcare system burdens and individual vaccination fatigue. Ye et al. introduced an mRNA-based combination vaccine, AR-CoV/IAV, which encodes the hemagglutinin (HA) protein of seasonal influenza A/H1N1 and the RBD of SARS-CoV-2, both encapsulated in lipid nanoparticles (LNPs) [248]. In a mouse model, this vaccine successfully elicited IgG and neutralizing antibody responses against both antigens, as well as robust CD4^+^ and CD8^+^ T cell responses. Huang et al. developed a Flu-COVID combo recombinant protein vaccine containing influenza HA and SARS-CoV-2 Spike (S) proteins adjuvanted with AddaVax, elicited protective immunity comparable to monovalent HA or S protein vaccines. The vaccine prevented body weight loss and clinical deterioration in K18-hACE2 mice challenged with lethal doses of influenza virus or SARS-CoV-2 [249]. Another study designed a fusion protein vaccine (H1N1 NP + SARS-CoV-2 RBD) leveraging preexisting influenza immunity to boost COVID-19 protection. Complete protection against morbidity and mortality and undetectable viral loads in lungs and nasal turbinates post-challenge in mice. Compared to influenza-naive mice, the vaccine enhanced RBD-specific antibody production in influenza-exposed mice [250].

Additionally, Moderna has developed several combination vaccines targeting RSV and other respiratory viruses. Among them, mRNA-1045 provides protection against seasonal influenza and RSV, while mRNA-1073 targets both SARS-CoV-2 and influenza. Furthermore, mRNA-1230 is designed as a trivalent vaccine to confer immunity against seasonal influenza, SARS-CoV-2, and RSV [251]. These advancements underscore the potential of combination vaccines to enhance disease prevention through a single-dose immunization strategy.

### 5.6. Multivalent Vaccines Expressing Multiple Viral Proteins

The emergence of highly transmissible Omicron variants, characterized by significant immune evasion, has posed challenges for vaccine strategies relying solely on the S protein. In contrast, the N protein, a highly conserved viral component across SARS-CoV-2 variants, is known to elicit robust T cell immunity. To enhance both neutralizing antibody production and broad protective T cell responses, several multivalent SARS-CoV-2 vaccines incorporating the S and N proteins have been developed, primarily utilizing viral vector platforms [252]. One example is hAd5 S-Fusion+N-ETSD, a vaccine that delivers SARS-CoV-2 S and N proteins via a human Ad5 platform. A combined IN and subcutaneous (SC) priming strategy demonstrated synergistic activation of both mucosal and systemic immunity in mice, leading to potent and durable neutralizing antibody and T cell responses [253]. Importantly, N-ETSD elicits CD4^+^ T cell responses, which are essential for inducing memory T cells and T helper cell-mediated B-cell antibody production. Additionally, a study on the combined administration of mRNA-S and mRNA-N vaccines reported enhanced protection in the upper respiratory tract and improved viral control in the lungs of hamster models challenged with Delta and Omicron variants [254]. In summary, the antigenic combination of S and N proteins offers a complementary immune response, where the S protein drives the breadth of neutralizing antibody production, while the N protein sustains T cell immunity through conserved epitopes, contributing to broader protection against SARS-CoV-2 variants.

### 5.7. Other Strategies to Optimize Neutralizing Antibodies

Apart from Fab-mediated interference with receptor binding, the Fc region of antibodies plays a crucial role in suppressing viral infection through effector functions such as antibody-dependent cellular cytotoxicity (ADCC) and antibody-dependent cellular phagocytosis (ADCP). These processes are mediated by immune cells, including macrophages and natural killer (NK) cells. For example, VIR-7831 (sotrovimab), an antibody engineered to enhance Fc effector functions based on S309, is one of the first cross-NAbs against SARS-CoV-2 to be used in clinical treatment [164]. Therefore, exploiting Fc effector functions can enhance the in vivo protective activity of antibodies.

A key challenge in antibody-based immunity is the glycan shielding of antigenic epitopes, which enables viral evasion from NAbs. Notably, N-glycosylation at the N370 site is conserved across 128 out of 129 sarbecovirus strains, except for SARS-CoV-2, where a T372A mutation leads to the loss of this modification [255]. The absence of this glycan alters antigen exposure, influencing immune recognition. Therefore, vaccine design must consider strategies to overcome glycan interference, allowing NAbs to achieve optimal recognition and efficacy.

Beyond overcoming immune evasion mechanisms, another promising approach to enhancing immune responses is the use of Toll-like receptor (TLR) agonists [256]. Studies have demonstrated that combinations of TLR4/TLR9 promote antigen cross-presentation and favor Th1-polarized IgG responses, while TLR4/TLR7/TLR9 combinations enhance antigen-specific antibody titers with a balanced Th1/Th2 response [256]. These findings underscore the potential of TLR agonist-based adjuvants in improving vaccine immunogenicity and facilitating the development of more effective SARS-CoV-2 vaccines.

Cryo-electron microscopy (cryo-EM) enables rapid atomic-level resolution of vaccine structures, significantly accelerating development efficiency and accuracy. This technique precisely characterizes key morphological parameters, including particle morphology, size distribution, and structural integrity, providing critical insights for rational vaccine design and optimization. Scripps Research scientists developed an accelerated vaccine development method using cryo-EM to rapidly characterize antibodies binding viral targets at atomic resolution. Their “structure-to-sequence” algorithm correlates cryo-EM-derived mAb structures with corresponding DNA sequences. In validation, screening 10^5^–10^6^ database sequences identified optimal matches to cryo-EM observed antibodies. Future advances in cryo-EM resolution and algorithmic efficiency could enable antibody identification solely through structural imaging, eliminating B-cell sequencing requirements [257].

Last but not least, artificial intelligence (AI) is emerging as a transformative tool in vaccine development. AI-driven algorithms such as LinearDesign enable the rapid optimization of mRNA vaccine stability and codon usage for the SARS-CoV-2 S protein [258]. This approach has been shown to significantly enhance mRNA half-life, improve protein expression, and increase antibody titers by up to 128-fold in mice. The integration of AI-based computational design into vaccine development holds immense potential for improving efficacy, scalability, and adaptability against emerging variants.

## 6. Future Perspectives

The development of COVID-19 vaccines presents both challenges and opportunities. While current vaccines have proven highly effective in preventing severe disease and reducing transmission, their efficacy against emerging variants, such as Omicron and its sublineages, has declined. Apart from SARS-CoV-2, MERS-CoV and SARS-CoV-1 are two other human coronaviruses (HCoVs) responsible for severe respiratory symptoms, and no vaccine is currently available for either. Furthermore, four endemic HCoVs—HCoV-229E, HCoV-OC43, HCoV-NL63, and HCoV-HKU1—have long circulated in human populations, primarily causing mild to moderate upper respiratory tract infections. Notably, frequent reinfections indicate that these endemic HCoVs fail to induce durable protective immunity. This highlights the need for continuous innovation in vaccine design to ensure broad and durable immunity against diverse HCoVs.

One promising direction is the development of next-generation vaccines targeting multiple variants or even different coronaviruses. These vaccines could leverage novel platforms, such as pan-coronavirus vaccines or mRNA vaccines, which allow for rapid adaptation to emerging strains. Moreover, increasing attention is being given to mucosal immunity, particularly in the upper respiratory tract. While most vaccines are administered intramuscularly to elicit systemic immunity, respiratory viruses like SARS-CoV-2 primarily enter through mucosal surfaces. Thus, stimulating a robust local immune response at the site of viral entry is crucial for preventing both infection and transmission. Mucosal vaccines have demonstrated enhanced efficacy in controlling respiratory infections in both animal models and clinical studies, largely due to their ability to induce local and systemic IgA, IgG, and T cell responses.

Furthermore, heterologous prime-boost regimens and multivalent vaccines may offer improved protection, particularly in populations with waning immunity. A combination of vaccines targeting both the S protein and other conserved regions, such as the N protein or the S2 subunit, could broaden immune protection and reduce the risk of immune escape among sarbecoviruses and even other β-coronaviruses. Meanwhile, the incorporation of novel adjuvants and advanced delivery systems may further enhance vaccine efficacy and durability.

In addition, small-molecule COVID-19 therapeutics have garnered significant interest from scientific and industrial communities due to distinct advantages, including simple administration, scalable production, and reduced immunogenicity risk. Over the past five years, multiple promising candidates demonstrating clinical efficacy have emerged, culminating in China’s National Medical Products Administration (NMPA) conditionally approving seven oral SARS-CoV-2 antivirals; these therapeutics primarily target viral inhibition or cytokine storm modulation, with SARS-CoV-2 inhibitors focusing predominantly on RdRp and 3CL^pro^ targets—specifically, RdRp inhibitors Azvudine, Molnupiravir, and Deuremidevir Hydrobromide, alongside 3CL^pro^ inhibitors Nirmatrelvir/Ritonavir, Simnotrelvir/Ritonavir, Leritrelvir, and Atilotrelvir/Ritonavir. Oral medications offer non-invasive, painless administration without requiring medical supervision. They are typically more cost-effective and demonstrate fewer adverse effects than alternative delivery routes like injections, while providing convenient options for repeated or long-term therapeutic use. Distinct pharmacokinetic profiles, polypharmacy risks, and teratogenic/developmental implications necessitate customized clinical approaches when administering medications to special populations such as older adults, chronic comorbid patients, cancer cases, pregnant persons, and children.

In conclusion, the future of COVID-19 vaccine development requires a multifaceted approach, including the refinement of existing vaccines, the exploration of novel technologies, and the strategic targeting of long-term immunity. Continuous adaptation and optimization of vaccine platforms will be essential to maintaining global health security against evolving viral threats.

## Figures and Tables

**Figure 1 vaccines-13-00635-f001:**
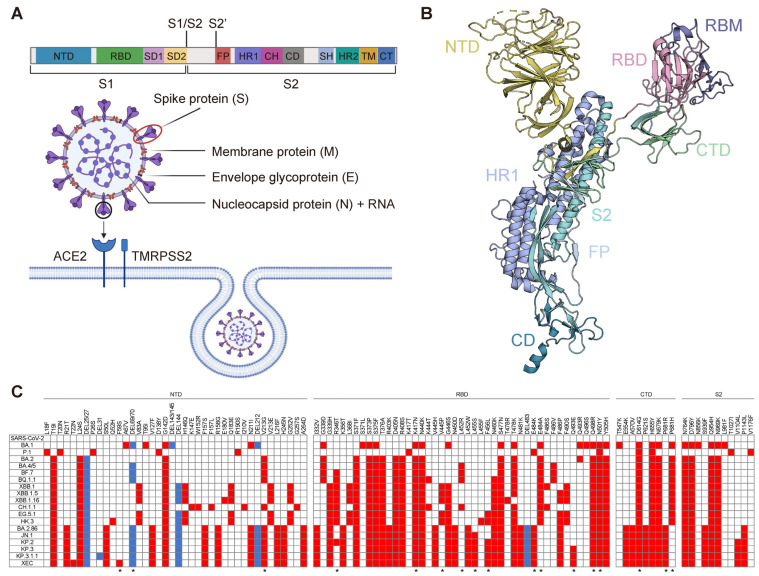
Structural organization of SARS-CoV-2 and progressive accumulation of key spike mutations in variants. (**A**) SARS-CoV-2 consists of four primary structural proteins: spike (S), nucleocapsid (N), membrane (M), and envelope (E). The S protein is subdivided into several functional domains, each indicated by a distinct color. Key regions include the S1 and S2 subunits, N-terminal domain (NTD), receptor-binding domain (RBD), S1/S2 and S2′ protease cleavage sites, fusion peptide (FP), heptad repeats (HR1 and HR2), central helix (CH), connector domain (CD), stem helix (SH), and the transmembrane (TM) domain. The schematic also illustrates the viral entry process into host cells. (**B**) Overall structure of the SARS-CoV-2 S trimer complex, highlighting major domains (PDB ID: 7DDN). (**C**) Progressive accumulation of representative S mutations in SARS-CoV-2 variants compared to the ancestral strain. Mutations are represented by color-coded bars: colored bars indicate presence, while white bars indicate absence. Asterisks mark key amino acid residues associated with immune escape, as identified in recent studies.

**Figure 2 vaccines-13-00635-f002:**
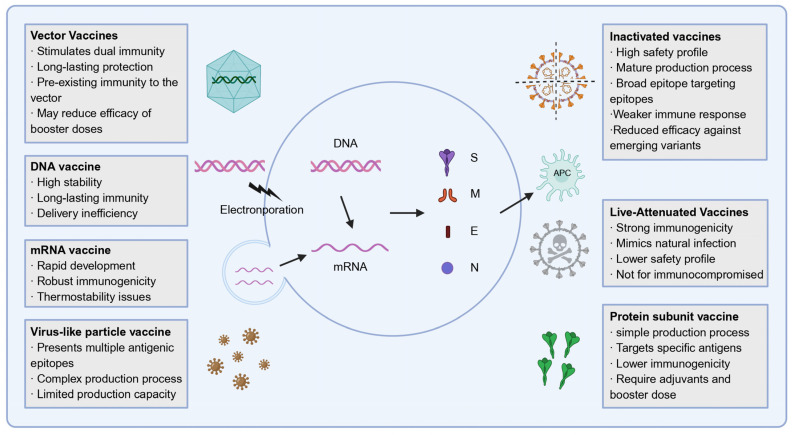
Overview of SARS-CoV-2 vaccine platforms and their associated immunogenic characteristics. This figure summarizes the major types of SARS-CoV-2 vaccines under development or in clinical use, categorized by platform: inactivated virus, protein subunit, viral vector-based (replicating and non-replicating), nucleic acid-based (mRNA and DNA), and virus-like particle (VLP) vaccines. Each vaccine type is depicted alongside its mechanism of action and characteristic immune responses, including humoral and cellular immunity profiles, as well as potential advantages and limitations.

**Figure 3 vaccines-13-00635-f003:**
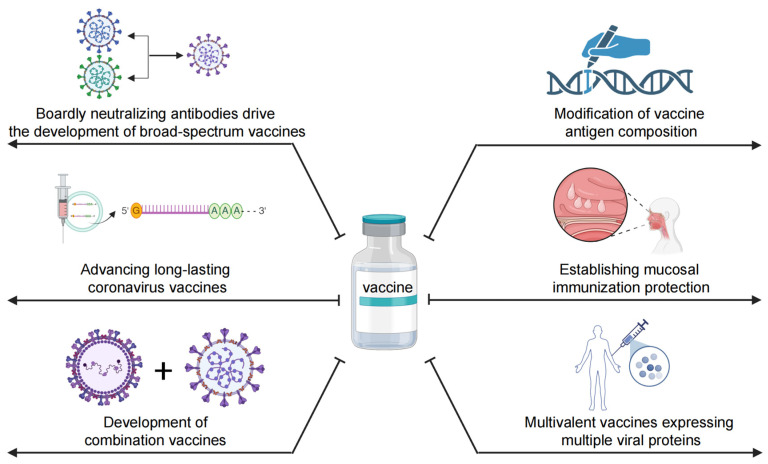
Potential strategies to optimize COVID-19 vaccines. This figure outlines multiple strategies aimed at enhancing the efficacy, breadth, and durability of COVID-19 vaccines. These approaches include antigen design optimization (e.g., inclusion of conserved regions such as the S2 domain or mosaic RBDs), use of adjuvants to boost immune responses, heterologous prime-boost regimens, and alternative delivery platforms such as intranasal or mucosal vaccines to induce local immunity. In addition, incorporating pan-sarbecovirus targets and updating immunogens to match emerging variants are highlighted as key directions for next-generation vaccine development.

**Table 2 vaccines-13-00635-t002:** Summary of SARS-CoV-2 antibodies.

No.	Name	Epitope	Detailed Information	Antibody Type	Developers	Article
1	Bamlanivimab	RBD	Isolated from PBMCs, obtained EUA in November 2020, restricted in April 2022 due to reduced Omicron efficacy	IgG	Chen P, et al.	[161]
2	Etesevimab		Isolated from PBMCs, obtained EUA in Feb 2021, modified CB6 enhanced neutralization of Omicron sublineages	IgG	Shi R, et al.	[162]
3	REGN10933/10987		Cocktail therapy with antibodies binding to RBD is ineffective against Omicron	IgG	Baum A, et al.	[163]
4	Sotrovimab		Derived from S309, targets a conserved epitope on the SARS-CoV-2 RBD	IgG	Pinto D, et al.	[164]
5	Tixagevimab		Combines with Cilgavimab enhances broad-spectrum neutralization against SARS-CoV-2 variants	IgG	Zost SJ, et al.	[165]
6	Cilgavimab		Has strong neutralization against SARS-CoV-2 variants, binds non-overlapping epitopes with Tixagevimab			
7	Bebtelovimab		Neutralizes SARS-CoV-2 variants (Omicron, BA.2, Delta), binds key RBD residues for ACE2 interaction	IgG	Westendorf K, et al.	[166]
8	Regdanvimab		Neutralizes SARS-CoV-2 and VOCs variants, targets key RBD overlapping ACE2 region	IgG	Kim C, et al.	[167]
9	CR9		Neutralizes SARS-CoV-2 and Omicron subvariants, inhibits viral replication	IgG	Chen Z, et al.	[168]
10	BRII-196		Targets SARS-CoV-2 RBD, blocking virus-ACE2 interaction	IgG	Ju B, et al.	[169]
11	BRII-198		Targets SARS-CoV-2 RBD, blocking virus-ACE2 interaction			
12	7F		Targets conserved RBD, neutralizes SARS-CoV-2, SARS-CoV-1, WIV16	VHH	Swart IC, et al.	[170]
13	RBD-chAb-45		Chimeric antibody targeting SARS-CoV-2 RBD, effectively neutralize SARS-CoV-2	Humanized mAb	Liang KH, et al.	[171]
14	C1596	NTD	Recognizes NTD epitope, binds multiple Omicron subvariants	IgG	Rubio AA, et al.	[172]
15	SARS2-57		Binds NTD loops, binds Alpha, Gamma, Delta, but not Beta, Omicron BA.1	Murine mAb	Adams LJ, et al.	[173]
16	C1717		Recognizes NTD and SD2 near viral membrane, neutralizes Beta, Gamma, and Omicron	IgG	Wang Z, et al.	[174]
17	BD58-0730/0771/ 0784/0786/0790		Bind a unique epitope on the N1/N2 loop of NTD, neutralize Omicron sub-lineages	IgG	Niu X, et al.	[175]
18	3711		Targets the silent face of the NTD, shows efficient neutralization against SARS-CoV-2	IgG	Zhang Z, et al.	[176]
19	NT-193		Binds NTD N3/N5 loops, neutralizes WA1, Beta, Alpha, Gamma and original Wuhan strain	Humanized mAb	Onodera T, et al.	[177]
20	WS6	S2	Targets conserved S2 epitope, neutralizes SARS-CoV-1 and SARS-CoV-2	Murine mAb	Shi W, et al.	[178]
21	CC40.8		Targets S2 stem helix, neutralizes SARS-CoV-2 variants and SARS-CoV-1	IgG	Zhou P, et al.	[179]
22	S2-4D/5D/8D		Recognizes a conserved epitope in the S2 subunit, neutralizes of diverse SARS-CoV-2 variants	Murine mAb	Li C, et al.	[180]
23	S2-4A		Inhibits membrane fusion, neutralizes SARS-CoV-2 variants, Targets residues such as E1144 and F1148			

This table summarizes key information on various SARS-CoV-2 antibodies, including antibody names, target epitopes, molecular characteristics, antibody types, and associated research and development institutions. These antibodies predominantly target specific viral epitopes, such as the RBD, NTD, or S2 subunit, and exhibit distinct neutralizing activities and mechanisms of action. Several antibodies have received emergency use authorization or demonstrated broad neutralizing efficacy against multiple SARS-CoV-2 variants in clinical studies, representing critical tools for therapeutic intervention and virological research.

**Table 3 vaccines-13-00635-t003:** Summary of pan-sarbecovirus antibodies.

No.	Name	Epitope	Detailed Information	Antibody Type	Developers	Article
1	M8a-3/-31/-34	RBD	Show cross-reactive against SARS-CoV-2 variants and animal sarbecoviruses	Humanized mAb	Fan C, et al.	[181]
2	SCM12-61/13-65 VSM9-12/44/8-83/16-12		IC_50_ values for 8 Sarbecoviruses in clade 1a and 1b range from 0.001 to 1.65 μg/mL	IgG	Hu Y, et al.	[182]
3	2-36		Neutralizes clade 1 and 2 sarbecovirus, IC_50_ values range from 0.002 to 0.658 μg/mL	IgG	Wang P, et al.	[183]
4	E7		Shows ultrapotent activity against all sarbecoviruses via quaternary structure	IgG	Chia WN, et al.	[184]
5	2-10, 2-31, 2-45, 2-67		Neutralization potency varies, with 2-67 showing reduced efficacy against clade 1b sarbecoviruses	VHH	Xiang Y, et al.	[185]
6	CYFN1006-1/1006-2		CYFN1006-1 neutralizes SARS-CoV-2 variants, SARS-CoV-1 and JN.1 subvariant; CYFN1006-2 neutralizes SARS-CoV-2 variants less effectively than SARS-CoV-1	IgG	Yu L, et al.	[186]
7	DH1047		Neutralizes SARS-CoV-2 2AA MA, SARS-CoV, bat coronaviruses WIV-1 and RsSHC014	IgG	Martinez DR, et al.	[187]
8	10-40		Broadly neutralizes clade 1 sarbecovirus, provides protection against SARS-CoV-2 and SARS-CoV	IgG	Liu L, et al.	[188]
9	PW5-4, 5-5, 5-535		show neutralizing activity against SARS-CoV-2 variants, SARS-CoV and clade 1 sarbecoviruses	IgG	Zhao X, et al.	[22]
10	Tnb04-01, Tnb03		Targets the SARS-CoV-2 RBD, showing potent activity via a conserved pocket	VHH	Dong H, et al.	[189]
11	6D6, 7D6		Target cryptic RBD site, cross-neutralize clade 1/2 and clade 3	Murine mAb	Li T, et al.	[190]
12	S2H97		Targets the core RBD and neutralizes GD-Pangolin, GX-Pangolin, WIV1 and SARS-CoV-1	IgG	Starr, T.N, et al.	[191]
13	TXG-0078	NTD	Binds N3/N5 loops of NTD, recognizes α/β-coronaviruses, shows broad binding and neutralization	IgG	Jonathan H, et al.	[192]
14	S2L28/S2M28/S2X333		Bind antigenic supersite on the pinnacle of the NTD, and neutralize SARS-CoV-2 and RaTG13	IgG	McCallum M, et al.	[193]
15	76E1	S2	Neutralizes SARS-CoV-2 and variants, cross-binds other human and γ/δ-coronaviruses	IgG	Sun X, et al.	[37]
16	CV3-25		Binds conserved stem-helix, neutralizes SARS CoV-1, SARS-CoV-2 and WIV1	IgG	Hurlburt NK, et al.	[194]
17	1249A8		Binds S2 domain (residues 1131–1171), neutralizes MERS-CoV, SARS CoV-1 and SARS-CoV-2	IgG	Piepenbrink MS, et al.	[195]
18	B6		Recognizes linear epitope in S2 stem helix, shows cross-reactivity from lineages A, B and C	Murine mAb	Sauer MM, et al.	[196]
19	S2P6		Targets stem helix, shows broad neutralization against sarbecoviruses, merbecovirus and embecovirus	IgG	Pinto D, et al.	[197]
20	28D9		Displayed cross-reactivity and reacted with merbecovirus, sarbecovirus and embecovirus	Humanized mAb	Wang C, et al.	[198]
21	COV44-62		Inhibits membrane fusion, neutralizes MERS-CoV, α-CoVs and β-CoVs	IgG	Dacon C, et al.	[38]
22	COV44-79		Inhibits membrane fusion, neutralizes α-CoVs and β-CoVs			

This table provides a summary of key information on pan-sarbecovirus antibodies, including antibody names, target epitopes, structural and functional characteristics, antibody types, and affiliated research institutions. These antibodies are capable of neutralizing diverse sarbecoviruses by targeting conserved epitopes within the RBD, NTD, or S2 subunit. Their broad-spectrum neutralizing activities highlight their therapeutic and prophylactic potential against current and emerging sarbecoviruses, offering valuable insights for the design of next-generation vaccines and antiviral strategies.

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
