# Peer review of "Evolving SARS-CoV-2 Vaccines: From Current Solutions to Broad-Spectrum Protection"

_vaccines, 2025, doi:10.3390/vaccines13060635_

Round 1

Reviewer 1 Report

Comments and Suggestions for Authors

The manuscript provides a comprehensive synthesis of SARS-CoV-2 vaccine development strategies, highlighting structural protein targets and diverse platforms ranging from inactivated vaccines to nucleic acid-based approaches. However, significant gaps in structural clarity and methodological depth limit its impact. For instance, sections on vaccine platforms (e.g., inactivated, viral vector, and mRNA vaccines) lack comparative analysis of their variant-specific efficacy. A more cohesive framework is needed to contrast platform-specific advantages—such as mRNA’s adaptability vs. viral vectors’ durability—against emerging variants like XBB.1.5 and JN.1. Furthermore, transitions between antigenic feature discussions (e.g., S protein domains) and vaccine design lack contextual bridges. A dedicated section linking conserved epitopes (e.g., S2 stem helix) to cross-reactive vaccine strategies would strengthen the narrative flow.

The conformational changes driven by NTD supersite glycosylation (e.g., XEC variant’s N234Q) and their impact on antibody accessibility remain unexplored. The T-cell immunity explanation section is brief. Expanding on conserved CD8+ epitopes in S2/N proteins and their role in cross-variant responses would provide a balanced immune perspective. The review also overlooks clinical real-world data, such as CDC reports on XBB.1.5 monovalent boosters reducing hospitalization by 52% in elderly cohorts, or 6-month durability of nasal vaccines like COVI-VAC in reducing transmission. The critical 2023–2025 advancements are omitted. The recent articles need to be included on conserved epitope vaccines (e.g., ferritin nanoparticles displaying RBD-S2 fusion proteins) and address regulatory challenges in combination vaccines (e.g., adjuvant compatibility in MVC-COV1901). Emphasizing actionable strategies—such as leveraging cryo-EM structural insights to refine mosaic antigens—would position the review as a forward-looking resource for both academic and industry stakeholders navigating the next phase of pandemic preparedness.

Figure 1’s S protein domain map should annotate key immune-evading mutations (e.g., BA.2.86’s V445P and JN.1’s L455S), while Table 1 needs columns for variant-specific neutralization data (e.g., fold-reduction in antibody titers against KP.3 compared to ancestral strains). Additionally, recent studies on FLiRT variants’ resistance to class I RBD antibodies and mucosal vaccines like iNCOVACC would enhance relevance. Revising Figure 2 to color-code platforms (e.g., green for viral vectors, orange for VLPs) and standardizing variant nomenclature (e.g., “BA.2.86” instead of “BA.2.86 variant”) would help readers. Minor issues include inconsistent terminology (e.g., alternating use of “broad-spectrum” and “pan-sarbecovirus” without definitions) and suboptimal figure resolution. Typographical errors (e.g., “photoxsidized” in Table 2) require correction.

Author Response

Evolving SARS-CoV-2 vaccines: from current solutions to broad-spectrum protection

vaccines-3647257

Rebuttal

We appreciate all the editor’s and reviewers’ supportive assessments of our work and their comments that are helpful in improving our manuscript. Meanwhile, we thank you for giving us the opportunity to revise and improve our manuscript. Below we respond to each reviewer’s points in detail, with notes as to where changes to the manuscript have been made. Please note that changes made in the text are marked in red text with yellow highlight.

Response to reviewers’ comments

Reviewer #1 (Comments to the Author):

The manuscript provides a comprehensive synthesis of SARS-CoV-2 vaccine development strategies, highlighting structural protein targets and diverse platforms ranging from inactivated vaccines to nucleic acid-based approaches. However, significant gaps in structural clarity and methodological depth limit its impact. For instance, sections on vaccine platforms (e.g., inactivated, viral vector, and mRNA vaccines) lack comparative analysis of their variant-specific efficacy. A more cohesive framework is needed to contrast platform-specific advantages—such as mRNA’s adaptability vs. viral vectors’ durability—against emerging variants like XBB.1.5 and JN.1.

Response: We extend our sincere gratitude for your thorough review and constructive feedback on our manuscript. Your observations are invaluable for enhancing the logical coherence of the paper. Given the journal’s word limit and to maintain focus on core arguments, we have refined the manuscript by optimizing existing phrasing and strengthening logical transitions, without altering the outline or adding new sections. Detail as follow:

  1. Added data on the booster dose effects of protein subunit vaccines WSK-V102C/D against emerging variants (JN.1.13, KP.2, KP.3) (Page 10);
  2. Included clinical evidence for XBB.1.5 mRNA boosters with analysis of immune escape mechanisms (Page 12);
  3. Expanded the analysis of vaccine effectiveness for the vector-based vaccine ChAdOx1-S (Page 13).

Furthermore, transitions between antigenic feature discussions (e.g., S protein domains) and vaccine design lack contextual bridges.

Response: We appreciate your valuable feedback and have added transition sentences preceding the vaccine design section (Page 6) as suggested. 

A dedicated section linking conserved epitopes (e.g., S2 stem helix) to cross-reactive vaccine strategies would strengthen the narrative flow.

Response: We appreciate your valuable suggestions and have incorporated text in the article to clarify the rationale for targeting the S2 region in broad-spectrum vaccines development (Page 19). 

The conformational changes driven by NTD supersite glycosylation (e.g., XEC variant’s N234Q) and their impact on antibody accessibility remain unexplored.

Response: We are deeply grateful for your evaluation and insightful comments. As noted, our review provided limited coverage of glycosylation sites. We have incorporated your suggestion by adding content on the critical role of NTD site glycosylation (specifically the T22N site in XEC variants) in shaping SARS-CoV-2 spike protein biology and immune evasion mechanisms, as recommended (Page 17).

The T-cell immunity explanation section is brief. Expanding on conserved CD8+ epitopes in S2/N proteins and their role in cross-variant responses would provide a balanced immune perspective.

Response: As acknowledged, T-cell immunity receives limited focus in our review. We appreciate your valuable insight regarding the significant protective effects of cross-reactive T cells in COVID-19. In response, we have expanded the discussion of their functions in the revised manuscript (Page 3). 

The review also overlooks clinical real-world data, such as CDC reports on XBB.1.5 monovalent boosters reducing hospitalization by 52% in elderly cohorts, or 6-month durability of nasal vaccines like COVI-VAC in reducing transmission.

Response: We appreciate your valuable suggestions and have incorporated clinical data discussions for relevant vaccine candidates as recommended (Page 12,13).

The critical 2023–2025 advancements are omitted. The recent articles need to be included on conserved epitope vaccines (e.g., ferritin nanoparticles displaying RBD-S2 fusion proteins) and address regulatory challenges in combination vaccines (e.g., adjuvant compatibility in MVC-COV1901).

Response: We sincerely appreciate your insightful comments. We have incorporated this discussion into the main text as suggested (Page 10,11).

Emphasizing actionable strategies—such as leveraging cryo-EM structural insights to refine mosaic antigens—would position the review as a forward-looking resource for both academic and industry stakeholders navigating the next phase of pandemic preparedness.

Response: We are grateful for your valuable input and have implemented the suggested discussion in the main text (Page 24).

Figure 1’s S protein domain map should annotate key immune-evading mutations (e.g., BA.2.86’s V445P and JN.1’s L455S), while Table 1 needs columns for variant-specific neutralization data (e.g., fold-reduction in antibody titers against KP.3 compared to ancestral strains). Revising Figure 2 to color-code platforms (e.g., green for viral vectors, orange for VLPs) and standardizing variant nomenclature (e.g., “BA.2.86” instead of “BA.2.86 variant”) would help readers.

Response: We sincerely appreciate your insightful comments. In response, we have added star markers to the heatmap in Figure 1C to highlight key mutation sites. Additionally, the manuscript text has been thoroughly reviewed to ensure consistency with standard variant nomenclature. We regret the omission of KP.3 neutralization data from Table 1, which is due to two primary factors: (1) by the time BA.1 emerged, most vaccines had already shown significantly reduced efficacy, making KP.3-specific assessments largely unexplored; and (2) comprehensive neutralization data for KP.3 remains incomplete at present.

Additionally, recent studies on FLiRT variants’ resistance to class I RBD antibodies and mucosal vaccine·s like iNCOVACC would enhance relevance.

Response: We sincerely appreciate your insightful guidance on providing crucial direction for enhancing our immune evasion section. We have incorporated a discussion of FLiRT variant mutations in the main text, with particular emphasis on how key mutation sites impact antibody binding (Page 17).

Minor issues include inconsistent terminology (e.g., alternating use of “broad-spectrum” and “pan-sarbecovirus” without definitions) and suboptimal figure resolution. Typographical errors (e.g., “photoxsidized” in Table 2) require correction.

Response: We sincerely appreciate your critical feedback regarding the precision of terminology. First, it is important to note that the definition of the term “broad-spectrum” varies depending on context and across researchers, and currently lacks a universally accepted standard. In this manuscript, we define “broad-spectrum” antibodies as those capable of neutralizing more than one virus. In contrast, the term “pan-sarbecovirus” specifically refers to antibodies that neutralize all SARS-CoV-2 subvariants as well as other sarbecoviruses utilizing ACE2 as the primary receptor. We have carefully reviewed the manuscript to correct any instances of imprecise usage or overstatement. Thank you for drawing our attention to these important issues.

Reviewer 2 Report

Comments and Suggestions for Authors

The manuscript provides a comprehensive review of vaccine targets and platforms developed so far for Covid-19. It also mentions the topic of therapeutic antibodies developed or under research that shifts the scope and flow of the article. It is recommeded that they expand the discussion to cover other therapeutic interventions as well, in order to provide a more balanced and comprehensive overview.

Author Response

Evolving SARS-CoV-2 vaccines: from current solutions to broad-spectrum protection

vaccines-3647257

Rebuttal

We appreciate all the editor’s and reviewers’ supportive assessments of our work and their comments that are helpful in improving our manuscript. Meanwhile, we thank you for giving us the opportunity to revise and improve our manuscript. Below we respond to each reviewer’s points in detail, with notes as to where changes to the manuscript have been made. Please note that changes made in the text are marked in red text with yellow highlight.

Response to reviewers’ comments

Reviewer #2 (Comments to the Author):

The manuscript provides a comprehensive review of vaccine targets and platforms developed so far for Covid-19. It also mentions the topic of therapeutic antibodies developed or under research that shifts the scope and flow of the article. It is recommeded that they expand the discussion to cover other therapeutic interventions as well, in order to provide a more balanced and comprehensive overview.

Response: Thank you for your valuable suggestion. We have comprehensively incorporated discussion of small-molecule therapeutics into the revised manuscript (Pages 23–24), while maintaining the original structure and preserving the manuscript’s core focus on vaccine development. We greatly appreciate your expert guidance.

Reviewer 3 Report

Comments and Suggestions for Authors

This is an extensive review of current vaccines against COVID-19, called SARS-CoV-2 vaccines in the article. The terminology reflects the choice of the authors to perform a theoretical review which does not consider the practical experience over 5 years in terms of protection against COVID-19 disease and deaths. It does not mention the safety experience with various vaccines either. While the article has its merits it should be complemented with short summaries of clinical experience with various types of vaccines.

Here are some examples

Whole virion vaccines were very important in the beginning of the pandemic, but have generally shown somewhat lower protection than e.g. mRNA vaccines. It is unlikely that these vaccines will be part of any future “solution”.

Adenovirus vectored vaccines have run into safety issues, such as blood clots in the brain, and are likely to be used less and less in the future.

mRNA vaccines targeting the S1 protein have been by far the most successful ones, so successful that any other category of COVID-19 vaccines can only be presented and discussed in relation to mRNA vaccines. A simple line listing like in the present paper is just not good enough.

The current problem with mRNA vaccine lies in the rapid mutation of the SARS-CoV-2 virus and the need for booster immunizations. It is not clear at all that “broad-spectrum” vaccines can help. This issue can of course be discussed, but the tone should be more critical. For comparison, “universal” influenza vaccines have been in the making for years but have never really succeeded to replace the current vaccines.

There is very little need to consider protection against other sarbecoviruses in this context. SARS-1 and MERS never gained pandemic dimensions. Mentions to these viruses should be deleted from the text and Abstract. Such a combination would not improve efficacy of COVID-19 vaccine.

Likewise, discussion of combination vaccines against other respiratory viruses is not relevant. While such combinations have been developed, they are not likely to improve protective efficacy against COVID-19.

After the above focusing (and deletions) the authors should present what is their favorite to increase protective efficacy and duration of protection of the current COVID-19 vaccines.

Author Response

Evolving SARS-CoV-2 vaccines: from current solutions to broad-spectrum protection

vaccines-3647257

Rebuttal

We appreciate all the editor’s and reviewers’ supportive assessments of our work and their comments that are helpful in improving our manuscript. Meanwhile, we thank you for giving us the opportunity to revise and improve our manuscript. Below we respond to each reviewer’s points in detail, with notes as to where changes to the manuscript have been made. Please note that changes made in the text are marked in red text with yellow highlight.

Response to reviewers’ comments

Reviewer #3 (Comments to the Author):

This is an extensive review of current vaccines against COVID-19, called SARS-CoV-2 vaccines in the article. The terminology reflects the choice of the authors to perform a theoretical review which does not consider the practical experience over 5 years in terms of protection against COVID-19 disease and deaths. It does not mention the safety experience with various vaccines either. While the article has its merits it should be complemented with short summaries of clinical experience with various types of vaccines.Here are some examples.

Response: We sincerely appreciate your profound insight on the significance of clinical experience. We fully acknowledge the critical value of real-world data over the past five years for vaccine evaluation and have carefully considered your suggestions regarding safety and protection efficacy. This review is intentionally focused on the technological mechanisms and theoretical foundations underlying vaccine development. Clinical translation and real-world evidence will be addressed in future dedicated studies. Thank you for your invaluable guidance.

Whole virion vaccines were very important in the beginning of the pandemic, but have generally shown somewhat lower protection than e.g. mRNA vaccines. It is unlikely that these vaccines will be part of any future “solution”.

Response: Thank you for your insightful comments regarding whole-virus vaccines. While we acknowledge the efficacy advantages of mRNA platforms, we have added content to provide a more balanced perspective by highlighting the critical role of whole-virus vaccines during the early phase of the pandemic, particularly their rapid deployment in resource-limited settings and their foundational efficacy, which in some cases exceeded 70% against the ancestral (WT) strain (see Page 6). Although future strategies may increasingly favor novel platforms, a comprehensive understanding of pandemic response necessitates recognition of the historical contributions of all vaccine technologies. We sincerely appreciate your valuable feedback.

Adenovirus vectored vaccines have run into safety issues, such as blood clots in the brain, and are likely to be used less and less in the future.

Response: We fully acknowledge your important point regarding the safety of adenovirus-vectored vaccines. While the manuscript discusses general safety considerations, the specific risk of thrombosis with thrombocytopenia syndrome (TTS) associated with ChAdOx1-S had not been explicitly addressed. In response to your comment, we have added relevant content on this issue to the Vector Vaccines section (see Page 12 of the revised manuscript). We appreciate your thoughtful feedback, which has helped to improve the comprehensiveness of our safety discussion.

mRNA vaccines targeting the S1 protein have been by far the most successful ones, so successful that any other category of COVID-19 vaccines can only be presented and discussed in relation to mRNA vaccines. A simple line listing like in the present paper is just not good enough.

Response: We sincerely appreciate your insightful comments. In response to your suggestions, we have incorporated robust clinical evidence highlighting the advantages of mRNA vaccines, particularly their rapid development timelines and high efficacy rates (see Pages 11–12). We also affirm the significant role these vaccines played in containing the spread of COVID-19 during the pandemic.

The current problem with mRNA vaccine lies in the rapid mutation of the SARS-CoV-2 virus and the need for booster immunizations. It is not clear at all that “broad-spectrum” vaccines can help. This issue can of course be discussed, but the tone should be more critical. For comparison, “universal” influenza vaccines have been in the making for years but have never really succeeded to replace the current vaccines.

Response: We are deeply grateful for your evaluation and insightful comments, particularly regarding the pragmatic limitations of broad-spectrum vaccines. In direct response to your comment, we have adopted a more critical stance regarding discussions of "broad-spectrum vaccines" due to unresolved scientific challenges (Page 19). Current limitations include substantial antigenic variation among coronaviruses and unvalidated cross-protection claims. Future vaccine development should prioritize epitope-focused strategies with robust preclinical validation before advancing to clinical trials.

There is very little need to consider protection against other sarbecoviruses in this context. SARS-1 and MERS never gained pandemic dimensions. Mentions to these viruses should be deleted from the text and Abstract. Such a combination would not improve efficacy of COVID-19 vaccine.

Response: We sincerely appreciate your expert guidance on refining the scope of our manuscript. Among the numerous respiratory virus threats to humans, SARS-CoV-1, MERS-CoV and SARS-CoV-2 can trigger severe respiratory symptoms in humans. In recent years, a succession of broad-spectrum antibodies targeting these three coronaviruses have emerged. This phenomenon strongly suggests the presence of a certain level of genetic conservation among these coronaviruses. Nevertheless, it is essential to critically assess the complexity underlying this trend. In this review, the inclusion of MERS-CoV is primarily intended to enrich the background information of the discussion and serve as a logical link in the scientific reasoning. There is no in-depth exploration of its vaccine protection mechanism and practical application.

While genetic conservation among coronaviruses provides a theoretical basis for broad-spectrum strategies, the epidemiological reality must be acknowledged: neither SARS-1 nor MERS has caused a global pandemic comparable to SARS-CoV-2. Given this distinction, all vaccine designs presented herein exclusively target SARS-CoV-2.

Currently, SARS-CoV-2 persistent mutations critically compromise vaccine efficacy and directly impact global pandemic control. Although cross-coronavirus protection holds long-term scientific value, current research imperatives and practical needs position it as a future objective. Subsequent studies should deepen our understanding of diverse coronaviruses and leverage advancing biotechnologies to establish foundations for this goal.

Likewise, discussion of combination vaccines against other respiratory viruses is not relevant. While such combinations have been developed, they are not likely to improve protective efficacy against COVID-19.

Response: We appreciate your perspective on combination vaccines. SARS-CoV-2 and influenza viruses are highly transmissible airborne pathogens associated with significant morbidity and mortality. Co-infection with both viruses has caused severe disease during recent cocirculation periods. Vaccination remains the most effective preventive measure against these diseases. However, current influenza and COVID-19 vaccines utilize distinct manufacturing platforms, requiring individuals to receive separate administrations for comprehensive protection. While combination vaccines may not enhance COVID-19 protection efficacy, they represent an emerging trend addressing pandemic control needs.

Huang et al. developed a Flu-COVID combo recombinant protein vaccine containing influenza HA and SARS-CoV-2 Spike proteins adjuvanted with AddaVax, elicited protective immunity comparable to monovalent HA or S protein vaccines. The vaccine prevented body weight loss and clinical deterioration in K18-hACE2 mice challenged with lethal doses of influenza virus or SARS-CoV-2. Other study designed a fusion protein vaccine (H1N1 NP + SARS-CoV-2 RBD) lever-aging preexisting influenza immunity to boost COVID-19 protection. Complete protection against morbidity and mortality and undetectable viral loads in lungs and nasal turbinates post-challenge in mice. Compared to influenza-naive mice, the vaccine enhanced RBD-specific antibody production in influenza-exposed mice

Furthermore, the simultaneous circulation of influenza and COVID-19 increases risks of vaccination errors, as individuals needing COVID-19 vaccination may have recently received influenza vaccines, or vice versa. In addition, separate administrations of both vaccines are required to prevent dual infection, increasing healthcare system burdens and individual vaccination fatigue. We appreciate the suggestion for a combination vaccine again. This recommendation has been incorporated into the main text of the manuscript (Page 21,22).

After the above focusing (and deletions) the authors should present what is their favorite to increase protective efficacy and duration of protection of the current COVID-19 vaccines.

Response: We have revised the manuscript per your valuable suggestions, and we sincerely appreciate your insightful expertise, which has greatly contributed to improving the scientific rigor and logical coherence of this work.

Round 2

Reviewer 1 Report

Comments and Suggestions for Authors

The responses to major comments are satisfactorily fulfilled.